# Adaptive dynamics of memory-one strategies in the repeated donation game

**Philip LaPorte**[1]*, **Christian Hilbe**[2]*, **Martin A. Nowak**[3,4]

**1** Department of Mathematics, University of California, Berkeley, Berkeley, California, United States of America, **2** Max Planck Research Group 'Dynamics of Social Behavior', Max Planck Institute for Evolutionary Biology, Plön, Germany, **3** Department of Mathematics, Harvard University, Cambridge, Massachussetts, United States of America, **4** Department of Organismic and Evolutionary Biology, Harvard University, Cambridge, Massachussetts, United States of America

\* philip_laporte@berkeley.edu (PL); hilbe@evolbio.mpg.de (CH)

## Abstract

Human interactions can take the form of social dilemmas: collectively, people fare best if all cooperate but each individual is tempted to free ride. Social dilemmas can be resolved when individuals interact repeatedly. Repetition allows them to adopt reciprocal strategies which incentivize cooperation. The most basic model for direct reciprocity is the repeated donation game, a variant of the prisoner's dilemma. Two players interact over many rounds; in each round they decide whether to cooperate or to defect. Strategies take into account the history of the play. Memory-one strategies depend only on the previous round. Even though they are among the most elementary strategies of direct reciprocity, their evolutionary dynamics has been difficult to study analytically. As a result, much previous work has relied on simulations. Here, we derive and analyze their adaptive dynamics. We show that the four-dimensional space of memory-one strategies has an invariant three-dimensional subspace, generated by the memory-one counting strategies. Counting strategies record how many players cooperated in the previous round, without considering who cooperated. We give a partial characterization of adaptive dynamics for memory-one strategies and a full characterization for memory-one counting strategies.

## Author summary

Direct reciprocity is a mechanism for evolution of cooperation based on the repeated interaction of the same players. In the most basic setting, we consider a game between two players and in each round they choose between cooperation and defection. Hence, there are four possible outcomes: (i) both cooperate; (ii) I cooperate, you defect; (ii) I defect, you cooperate; (iv) both defect. A memory-one strategy for playing this game is characterized by four quantities which specify the probabilities to cooperate in the next round depending on the outcome of the current round. We study evolutionary dynamics in the space of all memory-one strategies. We assume that mutant strategies are generated in

**Data Availability Statement:** All relevant data are within the manuscript.

**Funding:** C.H. acknowledges generous support by the European Research Council Starting grant 850529: E-DIRECT. The funders had no role in

study design, data collection and analysis, decision to publish, or preparation of the manuscript.

**Competing interests:** The authors have declared that no competing interests exist.

close proximity to the existing strategies, and therefore we can use the framework of adaptive dynamics, which is deterministic.

## Introduction

Evolution of cooperation is of considerable interest, because it demonstrates that natural selection does not only lead to selfish, brutish behavior red in tooth and claw [1, 2]. Yet in absence of a mechanism for its evolution, natural selection opposes cooperation. A mechanism for evolution of cooperation is an interaction structure that allows natural selection to favor cooperation over defection [3]. Direct reciprocity is one such mechanism [4–8]. This mechanism is based on repeated interactions among the same individuals. In a repeated interaction, individuals can condition their decisions on their co-player's previous behavior. By being more cooperative towards other cooperators, they can generate a favorable social environment for the evolution of cooperation.

The most basic model to illustrate reciprocity is the repeated donation game [1]. This game takes place among two players, who interact for many rounds. Each round, players independently decide whether to cooperate or defect. Cooperation implies a cost $c$ for the donor and generates a benefit $b$ for the recipient. Defection implies no cost and confers no benefit. Both players decide simultaneously. If they both cooperate, each of them gets payoff $b - c$. If both players defect, each of them gets payoff 0. If one player cooperates while the other defects, the cooperator's payoff is $-c$ while the defector's is $b$. The donation game is a special case of a prisoner's dilemma if $b > c > 0$, which is normally assumed.

If the donation game is played for a single round, players can only choose between the two possible strategies of cooperation and defection. Based on the game's payoffs, each player prefers to defect, creating the dilemma. In contrast, in the repeated donation game, infinitely many strategies are available. For example, players may choose to cooperate if and only if their co-player cooperated in the previous round. This is the well-known strategy Tit-for-tat [5, 9]. Alternatively, players may wish to occasionally forgive a defecting opponent, as captured by Generous Tit-for-tat [10, 11]. Against each of these strategies, unconditional defection is no longer the best response. Instead, mutual cooperation is now in the co-player's best interest.

During the past decades, there has been a considerable effort to explore whether conditionally cooperative behaviors would emerge naturally (e.g., [12–24]). To this end, researchers study the dynamics in evolving populations, in which strategies are transmitted either by biological or cultural evolution (by inheritance or imitation). For such an analysis, it is useful to restrict the space of strategies that individuals can choose from. The strategy space ought to be small enough for a systematic analysis, yet large enough to capture the most interesting behaviors.

One frequently used subspace is the set of memory-one strategies [24–32]. Players with memory-one strategies respond to the outcome of the previous round only. Such strategies can be written as a vector $\mathbf{p} = (p_{CC}, p_{CD}, p_{DC}, p_{DD})$ in the 4-dimensional cube $[0, 1]^4$. Each entry $p_{ij}$ reflects the player's conditional cooperation probability, depending on the four possible outcomes of the previous round, $CC, CD, DC, DD$ (the first letter is the focal player's action, the second letter is the co-player's action). Despite their simplicity, memory-one strategies can capture many different behavioral archetypes. They include always defect, ALLD = (0, 0, 0, 0), always cooperate, ALLC = (1, 1, 1, 1), Tit-for-tat, TFT = (1, 0, 1, 0) [5, 9], Generous Tit-for-tat, GTFT = (1, $x$, 1, $x$) with $0 < x < 1$ [10, 11], and Win-stay, Lose-shift, WSLS = (1, 0, 0, 1) [25,

33]. The sixteen corner points of the cube are the pure strategies. The interior of the cube are stochastic strategies. The center of the cube is the random strategy (1/2, 1/2, 1/2, 1/2) [5].

Conditionally cooperative strategies have been of particular interest in the study of human behavior. For example, there is evidence for the intuitive expectation that people tend to cooperate more if their co-player was cooperative in the past, or if they expect their co-player to cooperate in the future [34–36]. The concept of conditionally cooperative strategies is quite broad and includes strategies such as Tit-for-two-tats, which cannot be realized as a memory-one strategy. In this paper we consider only conditionally cooperative strategies which can be realized as memory-one strategies, such as TFT, GTFT, and nearby strategies. However, it is hoped that techniques similar to the ones used in this paper can be used to study more general strategy spaces.

When both players adopt memory-one strategies, there is an explicit formula to derive their average payoffs (as described in the next section). Based on this formula, it is possible to characterize all Nash equilibria among the memory-one strategies [37–42]. In general, however the payoff formula yields a complex expression in the players' conditional cooperation probabilities $p_{ij}$. As a result, it is difficult to characterize the dynamics of evolving populations, in which players switch strategies depending on the payoffs they yield. Most previous work had to resort to individual-based simulations. Only in special cases, an analytical description has been feasible (for example, based on differential equations). One special case arises when individuals are restricted to use reactive strategies [43–48]. Reactive strategies only depend on the co-player's previous move. Within the memory-one strategies, they correspond to the 2-dimensional subset with $p_{CC} = p_{DC}$ and $p_{CD} = p_{DD}$. In addition, there has been work on the replicator dynamics among three strategies [15, 49], and on the dynamics among transformed memory-one strategies [50, 51]. Here, we wish to explore the dynamics among memory-one strategies directly, using adaptive dynamics [52, 53].

We begin by describing two interesting mathematical results. First, we show that under adaptive dynamics, the 4-dimensional space of memory-one strategies contains an invariant 3-dimensional subset. This subset comprises all "counting strategies". These strategies only depend on the number of cooperators in the previous round. They correspond to memory-one strategies with $p_{CD} = p_{DC}$. Second, we find that for the donation game, the adaptive dynamics exhibits an interesting symmetry between orbits forward-in-time and backward-in-time. We use these mathematical results to partially characterize the adaptive dynamics among memory-one strategies, and to fully characterize the dynamics among memory-one counting strategies.

## Model

We study the infinitely repeated donation game between two players. Each round, each player has the option to cooperate ($C$) or to defect ($D$). Players make their choices independently, not knowing their co-player's choice in that round. Payoffs in each round are given by the matrix

$$
\begin{array}{cc}
 & \begin{array}{cc} C & \quad D \end{array} \\
\begin{array}{c} C \\ D \end{array} & \begin{pmatrix} b-c & -c \\ b & 0 \end{pmatrix}
\end{array}
\tag{1}
$$

The entries correspond to the payoff of the row-player, with $b$ and $c$ being the benefit and cost of cooperation, respectively. We assume $b > c > 0$ throughout. The above payoff matrix is

a special case of a symmetric $2 \times 2$ game with matrix

$$
\begin{array}{cc}
 & \begin{array}{cc} C & \quad D \end{array} \\
\begin{array}{c} C \\ D \end{array} & \begin{pmatrix} R & S \\ T & P \end{pmatrix}
\end{array}
\tag{2}
$$

The payoff matrix (1) of the donation game satisfies the typical inequalities of a prisoner's dilemma, $T > R > P > S$ and $2R > T + S$. Moreover, it satisfies the condition of 'equal gains from switching',

$$
R + P = T + S
\tag{3}
$$

This condition ensures that if players interact repeatedly, their overall payoffs only depend on how often each player cooperates, independent of the timing of cooperation.

In the following we focus on repeated games among players with memory-one strategies. Each player's decision is determined by a four-tuple $\mathbf{p} = (p_{CC}, p_{CD}, p_{DC}, p_{DD})$. Depending on the outcome of the previous round, $CC$, $CD$, $DC$, or $DD$, the focal player responds by cooperating with probability $p_{CC}$, $p_{CD}$, $p_{DC}$, or $p_{DD}$, respectively.

Strategies with large $p_{CC}$ exhibit a high frequency of mutual cooperation and will receive relatively large payoffs in the donation game. We note that in games with other payoff matrices 2, it may be beneficial in the long run for players to take turns cooperating with each other while the other defects. This behavior is called $ST$-reciprocity, because players will alternately receive payoffs $S$ and $T$ rather than $R$ in every round. $ST$-reciprocity becomes superior to $R$-reciprocity in terms of payoffs when $S + T > 2R$, and it can be achieved by memory-one strategies such as $(p_1, 0, 1, p_4)$ with small but positive $p_1$, $p_4$. For an account of $ST$- and $R$-reciprocity in other $2 \times 2$ games such as the Chicken or Snowdrift game, see [54, 55]. For the donation game, where $S + T = R < 2R$, we are primarily interested in the evolution of mutual cooperation $CC$.

We refer to a memory-one strategy as a counting strategy if it satisfies $p_{CD} = p_{DC}$. A counting strategy only reacts to the *number* of cooperators in the previous round. If both players cooperated in the previous round, they cooperate with probability $p_{CC}$. If exactly one of the players cooperated, they cooperate with probability $p_{CD} = p_{DC}$, irrespective of whether the outcome was $CD$ or $DC$. If no one cooperated, the cooperation probability is $p_{DD}$. Memory-one counting strategies include all unconditional strategies (such as ALLC and ALLD), as well as the strategies GRIM = $(1, 0, 0, 0)$ and WSLS = $(1, 0, 0, 1)$.

If the two players employ memory-one strategies $\mathbf{p} = (p_{CC}, p_{CD}, p_{DC}, p_{DD})$ and $\mathbf{p}' = (p'_{CC}, p'_{CD}, p'_{DC}, p'_{DD})$, then their behavior generates a Markov chain with transition matrix

$$
M = \begin{pmatrix}
p_{CC}\,p'_{CC} & p_{CC}(1-p'_{CC}) & (1-p_{CC})p'_{CC} & (1-p_{CC})(1-p'_{CC}) \\
p_{CD}\,p'_{DC} & p_{CD}(1-p'_{DC}) & (1-p_{CD})p'_{DC} & (1-p_{CD})(1-p'_{DC}) \\
p_{DC}\,p'_{CD} & p_{DC}(1-p'_{CD}) & (1-p_{DC})p'_{CD} & (1-p_{DC})(1-p'_{CD}) \\
p_{DD}\,p'_{DD} & p_{DD}(1-p'_{DD}) & (1-p_{DD})p'_{DD} & (1-p_{DD})(1-p'_{DD})
\end{pmatrix}
\tag{4}
$$

That is, if $\mathbf{s}(n) = (s_{CC}(n), s_{CD}(n), s_{DC}(n), s_{DD}(n))$, and $s_{ij}(n)$ is the probability that the $\mathbf{p}$-player chooses $i$ and the $\mathbf{p}'$-player chooses $j$ in round $n$, then $\mathbf{s}(n + 1) = \mathbf{s}(n)M$. For $\mathbf{p}, \mathbf{p}' \in (0, 1)^4$, the Markov chain has a unique invariant distribution $\mathbf{v} = (v_{CC}, v_{CD}, v_{DC}, v_{DD})$. This distribution $\mathbf{v}$ corresponds to the left eigenvector of $M$ with respect to the eigenvalue 1, normalized such that the entries of $\mathbf{v}$ sum up to one. The entries of $\mathbf{v}$ can be interpreted as the average frequency of the four possible outcomes over the course of the game. Therefore we can

define the repeated-game payoff of the **p**-player as

$$A(\mathbf{p}, \mathbf{p}') = Rv_{CC} + Sv_{CD} + Tv_{DC} + Pv_{DD} \tag{5}$$

For a more explicit representation of the players' payoffs, one can use the determinant formula by [56], which is shown in Methods.

To explore how players adapt their strategies over time, we use adaptive dynamics [52, 53]. Adaptive dynamics is a method to study deterministic evolutionary dynamics in a continuous strategy space. The idea is that the population is (mostly) homogeneous at any given time. Mutations generate a small ensemble of possible invaders, which are very close to the resident in strategy space. These invaders can take over the population if they receive a higher payoff against the resident than the resident achieves against itself. In the limit of infinitesimally small variation between resident and invader, we obtain an ordinary differential equation. For memory-one strategies this differential equation takes the form

$$\dot{p}_{ij} = \left.\frac{\partial A(\mathbf{p}, \mathbf{p}')}{\partial p_{ij}}\right|_{\mathbf{p}=\mathbf{p}'} \qquad \text{with } i,j \in \{C, D\} \tag{6}$$

That is, populations evolve towards the direction of the payoff gradient. We derive an explicit representation of this differential equation in Methods. The resulting expression defines a flow on the cube $[0, 1]^4$. Our aim is to understand the properties of this flow.

## Results

### Structural properties of adaptive dynamics

We begin by describing two general properties of adaptive dynamics in the cube $[0, 1]^4$ of memory-one strategies. The first property concerns an invariance result. As we prove in Methods, the subspace of counting strategies is left invariant under adaptive dynamics. That is, if the initial population $\mathbf{p}(0)$ satisfies $p_{CD}(0) = p_{DC}(0)$ and $\mathbf{p}(t)$ is a solution of the dynamic (6), then $p_{CD}(t) = p_{DC}(t)$ for all times $t$. Therefore, if initially all population members only care about the number of cooperators, then the same is true for all future population members. This result does not require the specific payoffs of the donation game. Instead it is true for all symmetric $2 \times 2$ games. The result is useful because it allows us to decompose the space of memory-one strategies into three invariant sets: the set of strategies with $p_{CD} > p_{DC}$, with $p_{CD} = p_{DC}$, and with $p_{CD} < p_{DC}$. Each of these invariant subsets can be studied in isolation. In a subsequent section, we provide such an analysis for the counting strategies (with $p_{CD} = p_{DC}$) specifically.

As a second property, we observe an interesting symmetry between different orbits of adaptive dynamics. Specifically, if $(p_{CC}, p_{CD}, p_{DC}, p_{DD})(t)$ is a solution to (6) on some interval $t \in (a, b)$, then so is $(1 - p_{DD}, 1 - p_{DC}, 1 - p_{CD}, 1 - p_{CC})(-t)$ on the interval $t \in (-b, -a)$. This property implies that for every orbit forward in time, there is an associated orbit backward in time that exhibits the same dynamics. This result is specific to the donation games (or more precisely, to games with equal gains from switching). The formal proof of this symmetry is in Methods. In the following we provide an intuitive argument. To this end, consider the following series of transformations applied to the payoff matrix of a $2 \times 2$ game with equal gains

from switching ($R + P = S + T$):

$$
\begin{array}{cc} & \begin{array}{cc} C & D \end{array} \\ \begin{array}{c} C \\ D \end{array} & \begin{pmatrix} R & S \\ T & P \end{pmatrix} \end{array}
\xrightarrow[\text{payoff}]{\text{negating}}
\begin{array}{cc} & \begin{array}{cc} C & D \end{array} \\ \begin{array}{c} C \\ D \end{array} & \begin{pmatrix} -R & -S \\ -T & -P \end{pmatrix} \end{array}
$$

$$
\xrightarrow[\text{constant}]{\text{adding a}}
\begin{array}{cc} & \begin{array}{cc} C & D \end{array} \\ \begin{array}{c} C \\ D \end{array} & \begin{pmatrix} -R+(R+P) & -S+(S+T) \\ -T+(S+T) & -P+(R+P) \end{pmatrix} \end{array}
=
\begin{array}{cc} & \begin{array}{cc} C & D \end{array} \\ \begin{array}{c} C \\ D \end{array} & \begin{pmatrix} P & T \\ S & R \end{pmatrix} \end{array}
\quad (7)
$$

$$
\xrightarrow[\text{C and D}]{\text{exchanging}}
\begin{array}{cc} & \begin{array}{cc} C & D \end{array} \\ \begin{array}{c} C \\ D \end{array} & \begin{pmatrix} R & S \\ T & P \end{pmatrix} \end{array}
$$

Notice that we started and ended at the same game; this property is equivalent to equal gains from switching. But now it is easy to see that solutions to the associated ordinary differential equation transform correspondingly as follows,

$$
(p_{CC}, p_{CD}, p_{DC}, p_{DD})(t)
\xrightarrow[\text{payoff}]{\text{negating}}
(p_{CC}, p_{CD}, p_{DC}, p_{DD})(-t)
$$

$$
\xrightarrow[\text{constant}]{\text{adding a}}
(p_{CC}, p_{CD}, p_{DC}, p_{DD})(-t)
\quad (8)
$$

$$
\xrightarrow[\text{C and D}]{\text{exchanging}}
(1-p_{DD}, 1-p_{DC}, 1-p_{CD}, 1-p_{CC})(-t)
$$

The upshot of this duality is that solutions to adaptive dynamics come in related pairs. We will see expressions of this duality in several of the figures below.

## Adaptive dynamics of memory-one strategies

In the following, we aim to get a more qualitative understanding of the adaptive dynamics. To this end, we first examine which combinations of signs can appear in the components of the vector field $(\dot{p}_{CC}, \dot{p}_{CD}, \dot{p}_{DC}, \dot{p}_{DD})$. For example, it turns out that if $p_{CC}$ is decreasing, $p_{DC}$ must be decreasing as well. Similarly, if $p_{DD}$ is decreasing, then so is $p_{CD}$. For $c/b = 0.1$, the results of this sign analysis are shown in Fig 1. There we show a $9 \times 9 \times 9 \times 9$ evenly spaced grid on $[0, 1]^4$. Each point is colored according to the signs of the components of $\mathbf{\dot{p}} = (\dot{p}_{CC}, \dot{p}_{CD}, \dot{p}_{CD}, \dot{p}_{DD})$ at that point. Therefore, the figure provides information about the direction of adaptive dynamics at each point. We observe that the combinations $abcd$ of signs come in pairs of the form $abcd$, $dcba$. For example, there are exactly as many points having signs '$+---$' as '$---+$'. The sets of points in each pair are related to each other by reflection about the diagonal in the figure. If $abcd$ are the signs at $(x, y, z, w) \in (0, 1)^4$, then $dcba$ are the signs at $(1 - w, 1 - z, 1 - y, 1 - x)$. This is, of course, a consequence of the symmetry described in the previous section.

In a next step, we aim to find all interior fixed (critical) points of adaptive dynamics. As we show in Methods, these turn out to be the solutions to the linear system

$$
b(p_{CC}-p_{CD}) + c(-1+p_{CC}-p_{DC}) = 0 \quad \text{and} \quad p_{CC}+p_{DD} = p_{CD}+p_{DC} \quad (9)
$$

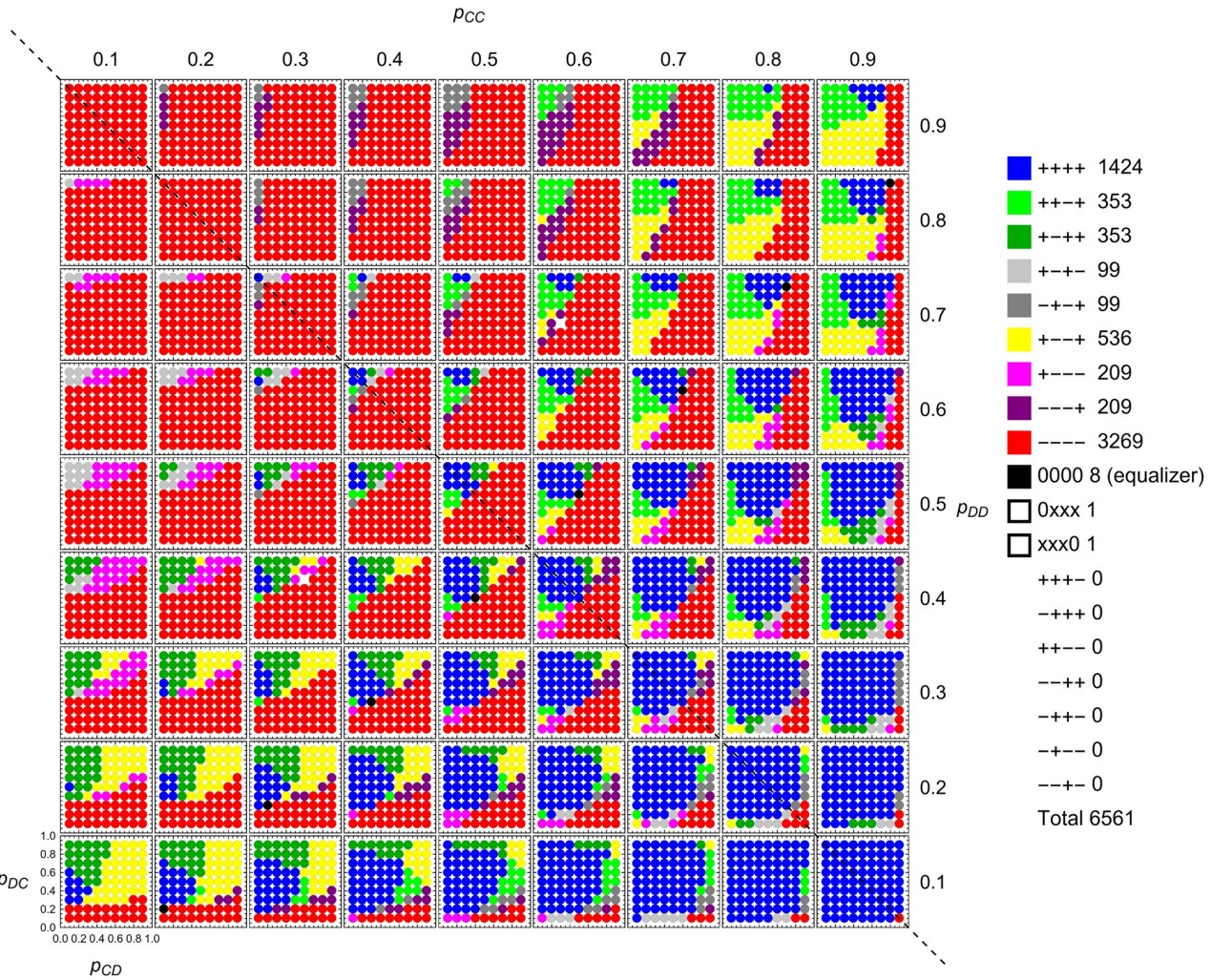

**Fig 1. Local adaptive dynamics for memory-one strategies.** For a $9 \times 9 \times 9 \times 9$-grid (= 6561 points) we show the direction of change in terms of the sign of each component of $(\dot{p}_{CC}, \dot{p}_{CD}, \dot{p}_{DC}, \dot{p}_{DD})$ as given by Eq (6). The possibilities are shown on the right. We observe that for 1424 points all four components are positive, $++++$. For 3269 points all four components are negative, $----$. Seven combinations do not occur. These combinations fall into one or both of the following categories: (*i*) $\dot{p}_{CC}$ is negative and $\dot{p}_{DC}$ is positive, and (*ii*) $\dot{p}_{DD}$ is negative and $\dot{p}_{CD}$ is positive. Both combinations are forbidden. Because of the symmetry (8) there are three pairs where each combination occurs as often as its partner. One such pair is $++-+$ and $+-++$ (each occurring 353 times). The configuration $+--+$ is its own mirror image and therefore a singleton (occurring 536 times). The reason for the symmetry in the plot is explained in the main text. Let $\sigma: [0, 1]^4 \rightarrow [0, 1]^4$ be defined by $\sigma(p_{CC}, p_{CD}, p_{DC}, p_{DD}) = (1-p_{DD}, 1-p_{DC}, 1-p_{CD}, 1-p_{CC})$. If *abcd* are the signs at **p**, then *dcba* are the signs at $\sigma(\mathbf{p})$. $\sigma$ acts by reflection about the dotted diagonal line shown. Finally, eight points are critical points with $(\dot{p}_{CC}, \dot{p}_{CD}, \dot{p}_{DC}, \dot{p}_{DD}) = (0, 0, 0, 0)$. Two points are zero in one but not all of the four components. The graph is created for $c = 0.1$.

In particular, the set of interior critical points forms a two-dimensional plane within the four-dimensional cube. As we will show in Methods, (9) implies certain bounds on $p_{CC}$ and $p_{DD}$ among the interior critical points: $p_{CC} > c/b$ and $p_{DD} < 1-c/b$.

By definition, critical points satisfy a local condition, $\dot{p}_{ij} = 0$ for all $i, j \in \{C, D\}$. However, it turns out that the critical points identified above have a shared global property. The points that satisfy (9) coincide with the equalizer strategies that have been described earlier [56, 57]. An equalizer is a strategy **p** such that $A(\mathbf{p}', \mathbf{p})$ is a constant, irrespective of $\mathbf{p}'$. Every such strategy must be a critical point of adaptive dynamics. Our result shows that also the converse is true. Every interior critical point of the system (6) needs to be an equalizer.

We can also examine what happens on the boundary of the strategy space. For our analysis, we define the boundary $\mathcal{B}([0, 1]^4)$ to be all points $\mathbf{p} \in [0, 1]^4$ with exactly one entry $p_{ij} \in \{0, 1\}$. That is, we exclude corner and edge points. What remains is a set of eight 3-dimensional cubes. We call a point $p \in \mathcal{B}([0, 1]^4)$ saturated if $p_{ij} = 0$ implies $\dot{p}_{ij} \leq 0$ and $p_{ij} = 1$ implies $\dot{p}_{ij} \geq 0$. A point is called strictly saturated if the above inequalities are strict. A point is unsaturated if it is not saturated. Orbits that start at an unsaturated point move into the interior of the strategy space. Conversely, every strictly saturated point is the limit, forward in time, of some trajectory in the interior.

For memory-one strategies, all eight boundary faces contain both saturated and unsaturated points for some values of $0 < c < b$ (Fig 2). In the following, we discuss in more detail the boundary face for which mutual cooperation is absorbing (that is, the boundary face with $p_{CC} = 1$). On this boundary face, the population obtains the socially optimal payoff of $b - c$, irrespective of the specific values of $p_{CD}, p_{DC}, p_{DD}$. As a result, we show in Methods that the time derivatives with respect to these components vanish, $\dot{p}_{CD} = \dot{p}_{DC} = \dot{p}_{DD} = 0$. The saturated points on the face $p_{CC} = 1$ are exactly those that satisfy $\dot{p}_{CC} \geq 0$, which yields the condition

$$\frac{(1-p_{CD})(1 - (1-p_{DC})(p_{CD}-p_{DD}) - (p_{DC}-p_{DD})^2)}{(1-p_{CD})^2(1-p_{DC}) + p_{DC}p_{DD}(2-p_{DD}) + (1-p_{CD})(1-p_{DC})(p_{DC}+p_{DD})} \geq \frac{c}{b} \tag{10}$$

This set of saturated points contains all cooperative memory-one Nash equilibria, which has been characterized by [38] to be the set of all strategies $\mathbf{p}$ that satisfy $p_{CC} = 1$ and

$$\frac{1-p_{CD}}{p_{DD}} \geq \frac{c}{b-c} \quad \text{and} \quad \frac{1-p_{CD}}{p_{DC}} \geq \frac{c}{b} \tag{11}$$

We note, that the conditions (11) are more strict than the conditions (10). Put another way, a boundary point can be a local maximum of the payoff function against itself without being a global maximum.

In a similar way, one can also characterize the saturated points on the boundary face with $p_{DD} = 0$, where mutual defection is absorbing. We depict the set of saturated points on this face in the bottom row of Fig 2, together with the previously discussed set of saturated points with $p_{CC} = 1$ in the top row. As the figure suggests, the two sets exactly complement each other. For every point that is strictly saturated on the boundary face $p_{CC} = 1$ there is a corresponding point on the face $p_{DD} = 0$ that is unsaturated. Of course, that correspondence is again a consequence of the symmetry described earlier.

After describing the critical points in the interior, and the saturated points on the boundary, we explore the 'typical' behavior of interior trajectories. To this end, we record the end behavior of solutions $\mathbf{p}(t)$ to Eq (6) beginning at various initial conditions $\mathbf{p}(0)$. Dynamics are assumed to cease at the boundary of the strategy space. This behavior can be numerically calculated. The results, for a $9 \times 9 \times 9 \times 9$ grid of initial conditions and cost-to-benefit ratio $c/b = 0.1$, are shown in Fig 3. There are 6561 initial conditions. Out of those, 1835 points are observed to end at full cooperation ($p_{CC} = 1$), 1375 points at full defection ($p_{DD} = 0$), 2964 points at other places on the boundary, and 387 at interior critical points (equalizers). Unlike in Fig 1, we do not observe the symmetry described in Eqs (7 and 8). The choice of depicting the forward direction of time breaks the symmetry.

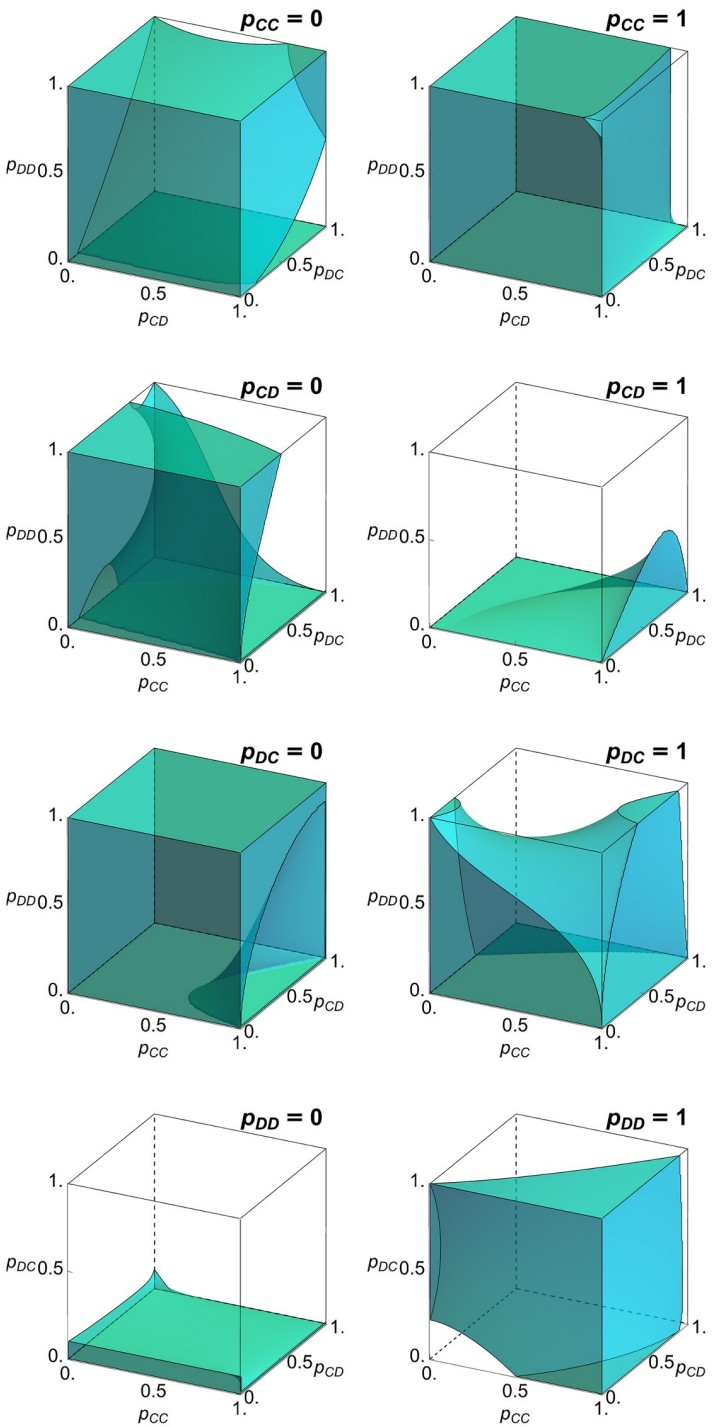

**Fig 2. Saturated points on the boundary of memory-one strategies.** The boundary of the set of memory-one strategies consists of eight three-dimensional faces with $p_{ij} = 0$ or $p_{ij} = 1$ for exactly one pair of $i, j \in \{C, D\}$. We omit points $(p_{CC}, p_{CD}, p_{DC}, p_{DD})$ for which more than one $p_{ij}$ is 0 or 1. Thus, the eight boundary faces do not intersect. A point **p** on the boundary is saturated if the payoff gradient does not point into the interior of the cube. We show the set of saturated points on all eight boundary faces. Because of the symmetry described by Eqs (7) and (8), these eight sets of points fit together in four complementary pairs, like the curved pieces of a three-dimensional puzzle. The boundary face $p_{ij} = 0$ is paired with the face $p_{\bar{i}\bar{j}} = 1$ (where a bar refers to the opposite action, $\bar{C} = D$ and $\bar{D} = C$). The paired boundary faces fit together after a rotation of one of them 180° about the line parameterized by $\left(t, \frac{1}{2}, 1 - t\right)$. Parameter $c = 0.1$.

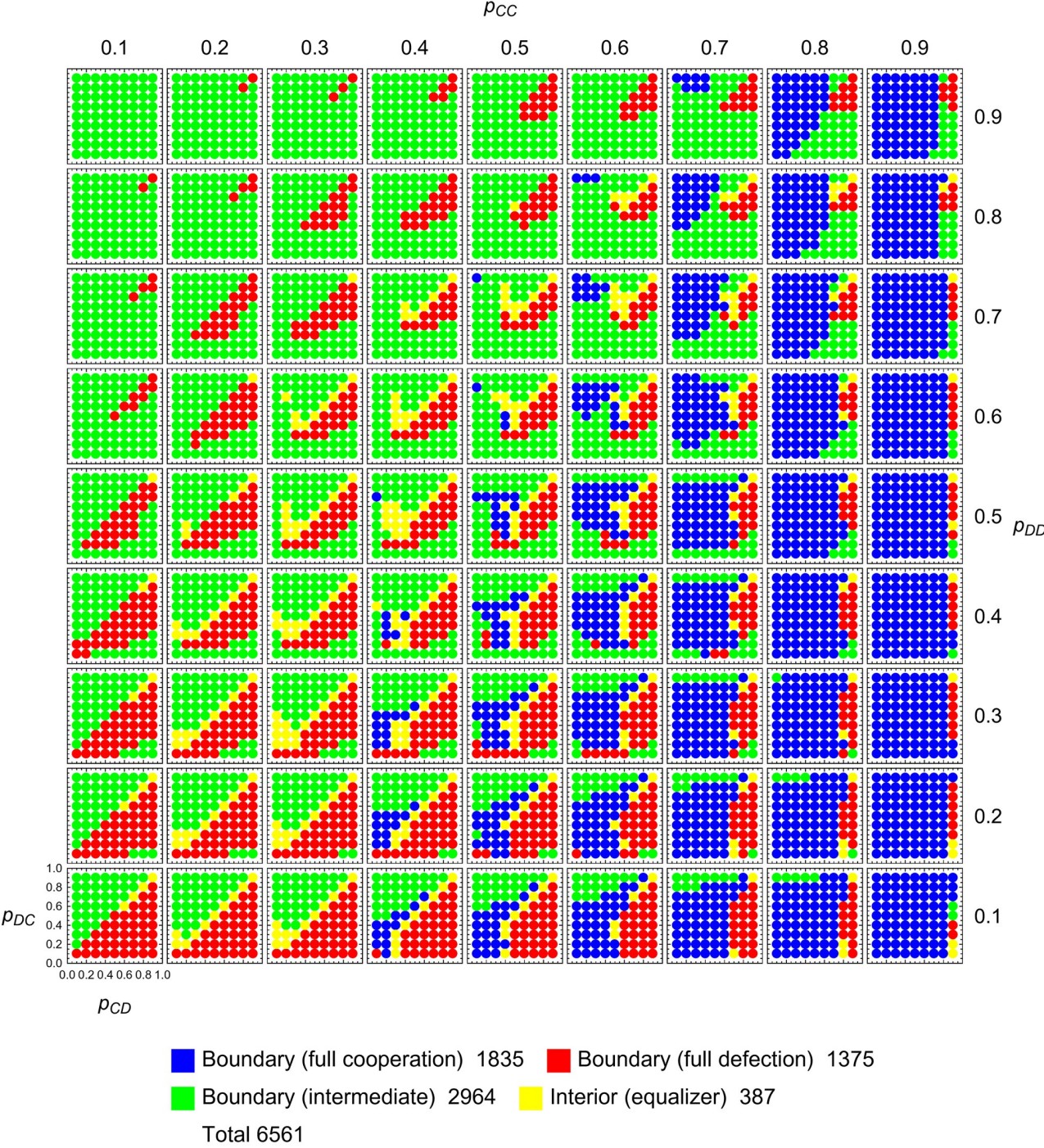

**Fig 3. Long-time limits of adaptive dynamics of memory-one strategies.** For a $9 \times 9 \times 9 \times 9$-grid of starting points (= 6561 points), we show the limit $\lim_{t \to \infty} \mathbf{p}(t)$ of a solution $\mathbf{p}(t)$ to Eq (6). Dynamics are assumed to cease at the boundary of the strategy space. Generically, there are 4 possibilities, as shown in the legend. For 1835 points, the trajectory $\mathbf{p}(t)$ evolves to full cooperation, defined by $p_{CC} = 1$ (blue). For 1375 points, the trajectory $\mathbf{p}(t)$ evolves to full defection, defined by $p_{DD} = 0$ (red). The remaining points either evolve into other regions of the boundary (green) or approach interior critical points, which are equalizers (yellow). The symmetry described in the main text does not manifest in this plot, but reappears when we juxtapose the plot with the corresponding plot for reversed time. Parameter $c = 0.1$.

## Adaptive dynamics of memory-one counting strategies

After describing the dynamics of memory-one strategies, we proceed by analyzing the dynamics of counting strategies, with $p_{CD} = p_{DC}$. Counting strategies are especially convenient because they can be represented in three dimensions. To make this representation explicit, in the following we write counting strategies as vectors $\mathbf{q} = (q_2, q_1, q_0) \in [0, 1]^3$. Here, $q_i$ is the probability to cooperate if $i$ of the two players have cooperated previously. The respective memory-one representation is thus given by $p_{CC} = q_2$, $p_{CD} = p_{DC} = q_1$, and $p_{DD} = q_0$. Correspondingly, the dynamics that we explore is given by

$$\dot{q}_i = \left.\frac{\partial A(\mathbf{q}, \mathbf{q}')}{\partial q_i}\right|_{\mathbf{q}=\mathbf{q}'} \qquad \text{with } i \in \{2, 1, 0\} \tag{12}$$

This dynamics among counting strategies is not identical to the previously considered dynamics among memory-one strategies, even when the starting population is taken from the invariant subset with $p_{CD} = p_{DC}$. Instead, differences arise because the embedding $[0, 1]^3 \rightarrow [0, 1]^4$ is not distance-preserving with the standard metric on each space. As a result, the gradient of the payoff function is computed slightly differently in the two spaces—specifically, the memory-one adaptive dynamics (6) within the subspace of counting strategies subspace differs from the adaptive dynamics (6) by a factor of 2 in $\dot{q}_1(t)$. The analysis in this section is thus not to characterize the orbits of the invariant subspace of counting strategies within the memory-one strategies. Rather we consider the space of counting strategies $[0, 1]^3$ as an interesting space in its own right, which we analyze in the following.

In a first step, we reproduce Fig 1 for the case of counting strategies. In Fig 1, counting strategies correspond to the points on the diagonal $p_{CD} = p_{DC}$ of each subpanel. Fig 4 is the analog of Fig 1 for counting strategies, where we plot the signs of the components of $(\dot{q}_2, \dot{q}_1, \dot{q}_0)$ at each counting strategy. As one may expect, these combinations again come in pairs, where $abc$ is paired with $cba$. Some combinations, such as +++, are self-paired.

Similar to the memory-one strategies, we also want to characterize the set of interior critical points of the system (12). In Methods, we show that these points can now be parametrized by

$$\left(t + \frac{c}{b+c}\ ,\ t\ ,\ t - \frac{c}{b+c}\right), \qquad \text{with}\ \ t \in \left(\frac{c}{b+c},\ \frac{b}{b+c}\right) \tag{13}$$

Hence the set of interior critical points forms a straight line segment. The boundary points of this line segment are

$$\left(\frac{2c}{b+c},\ \frac{c}{b+c}, 0\right) \qquad \text{and} \qquad \left(1,\ \frac{b}{b+c}, \frac{b-c}{b+c}\right) \tag{14}$$

The length of this line segment is $\sqrt{3}(b-c)/(b+c)$, which ranges from $\sqrt{3}$ (the diagonal of the cube) to 0, as $c/b$ ranges from 0 to 1. We can classify the stability of the critical points by finding their associated eigenvalues. The complete results are shown in Fig 5. Five generic types of critical points are present as we vary the cost-to-benefit ratio: source, spiral source, spiral sink, sink, and saddle.

In addition to these interior critical points, Fig 6 also depicts the critical points on the boundary faces $\mathcal{B}([0, 1]^3)$. Using the terminology of the previous section, these critical points are saturated without being strictly saturated. On each boundary face, the respective curve thus separates the region of strictly saturated points from the unsaturated points. Because of the aforementioned symmetry of solutions, the set of boundary critical points is symmetric under the transformation $(x, y, z) \mapsto (1 - z, 1 - y, 1 - x)$. We note that counting strategies have

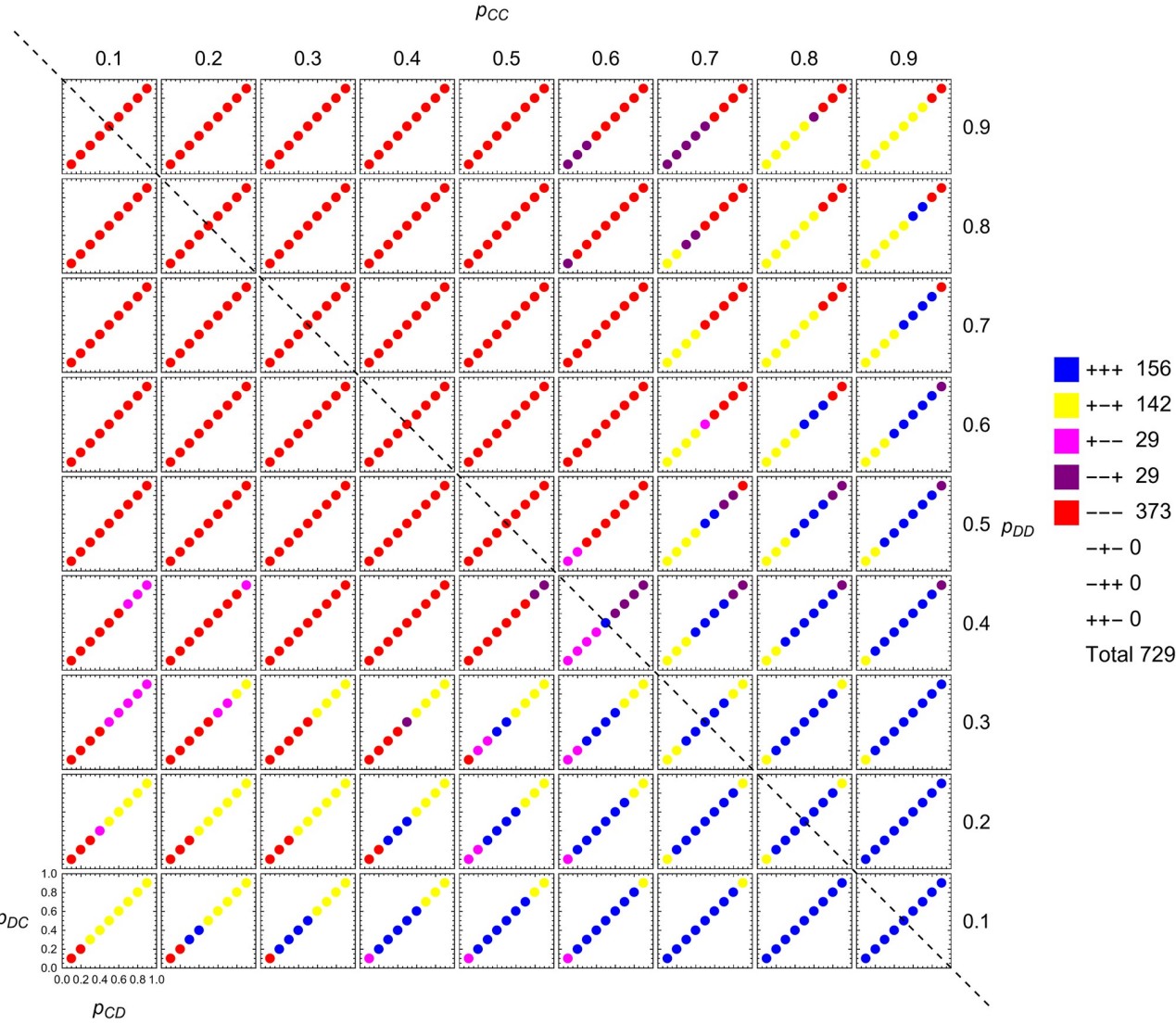

**Fig 4. Local adaptive dynamics for counting strategies.** On a $9 \times 9 \times 9 \times 9$-grid representing the space of memory-one strategies, we depict the 729 points which are counting strategies (defined by $p_{CD} = p_{DC}$). They are colored according to their direction of change in terms of the sign of each component of $(\dot{q}_2, \dot{q}_1, \dot{q}_0)$. Generically, there are eight possibilities as shown in the legend. We observe that for 156 points all three components are positive, +++, while for 373 points all three components are negative, ---. Three combinations do not occur: -+-, -++, and ++-. These are combinations in which $\dot{q}_2$ or $\dot{q}_0$ is negative while $\dot{q}_1$ is positive; such combinations are forbidden. Because of the symmetry derived in the main text there is a symmetric pair, +-- and --+, each occurring 29 times. The configuration +-+ is its own mirror image and therefore a singleton (occurring 142 times). Parameter $c = 0.1$.

boundary properties unshared by memory-one strategies. For example, every boundary point with $q_1 = 0$ is saturated. Conversely, every boundary point with $q_1 = 1$ is unsaturated.

To explore the dynamics in the interior, Fig 7 depicts the end behavior of solutions $\mathbf{q}(t)$ to Eq (12) with initial conditions on an evenly spaced grid (analogous to Fig 3). Again, dynamics are assumed to cease at the boundary. We observe that out of 729 initial points, 190 evolve to full cooperation, 140 evolve to full defection, 229 evolve to other places on the boundary, and 170 evolve to interior critical points. The overall abundance of the four outcomes is thus similar to the respective numbers in the space of all memory-one strategies, with the only exception being that now more orbits converge to interior critical points.

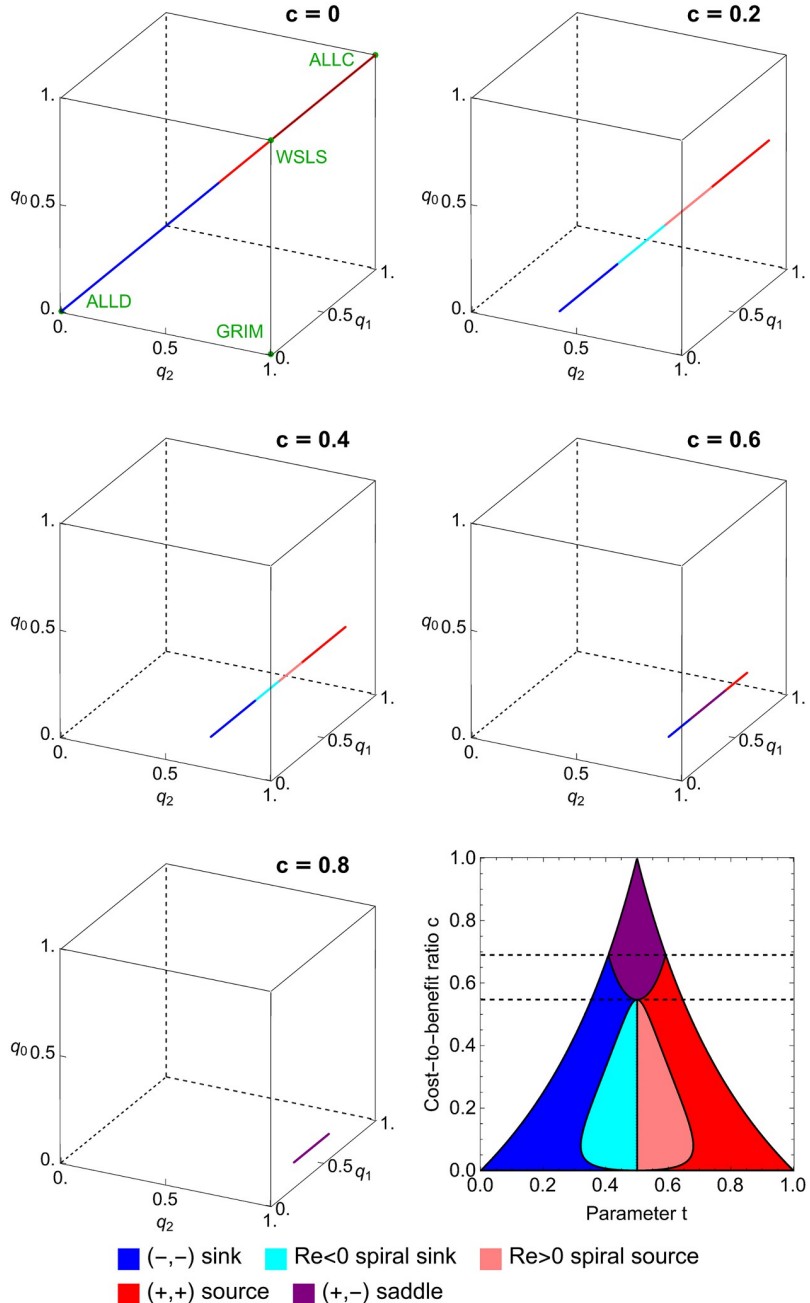

**Fig 5. Classification of interior critical points in the space of counting strategies.** We show the line of interior critical points in the space of counting strategies for five values of $c$. The line is colored according to the type of each critical point, which is determined by the eigenvalues of the linearization of the system (12) at this point. We observe all five generic types: source, spiral source, sink, spiral sink, and saddle. The complete classification is shown in the lower right panel. Each interior critical point is an equalizer (see main text). The line is parameterized by $(t + c/(1 + c), t, t - c/(1 + c))$ as $t$ ranges over the interval $(c/(1 + c), 1/(1 + c))$. The symmetry described in the main text is manifest in this figure. The transformation $\sigma$: $(x, y, z) \mapsto (1 - z, 1 - y, 1 - x)$ carries the line of critical points to itself. It exchanges sinks and sources, spiral sinks and spiral sources, and saddle points and other saddle points.

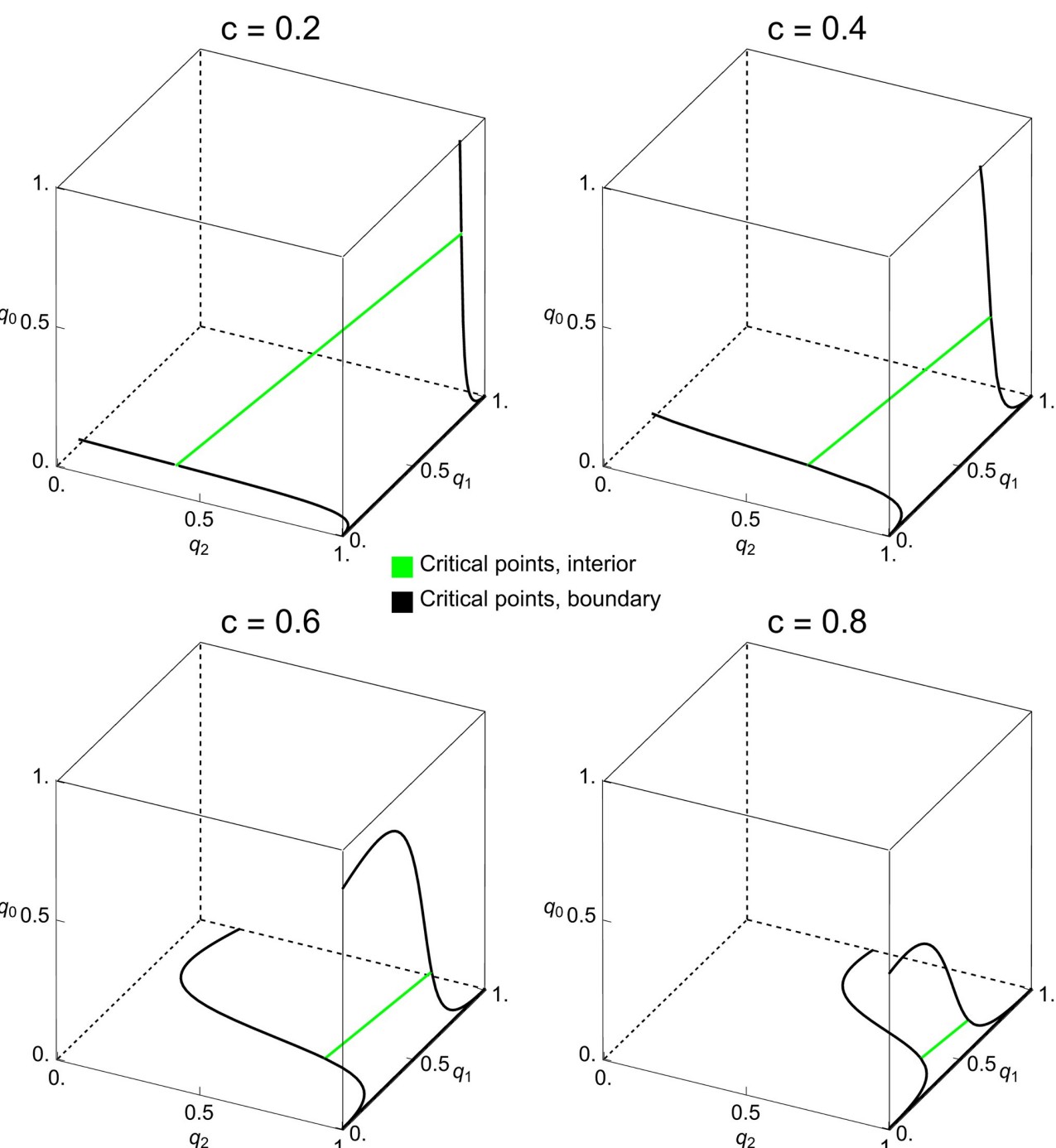

**Fig 6. Interior and boundary critical points in the space of counting strategies.** For four values of $c$, we show the line of interior critical points (green) and the boundary critical points (black) in the space of counting strategies. The boundary critical points consist of three pieces: the edge defined by $q_0 = 0$ and $q_2 = 1$ (i.e. the intersection of full cooperation and full defection) and two separate curves on the faces $q_0 = 0$ and $q_2 = 1$. For example, the strategy GRIM = $(1, 0, 0)$ is a boundary critical point. The symmetry described in the main text is visible in the rotational symmetry of the set of critical points.

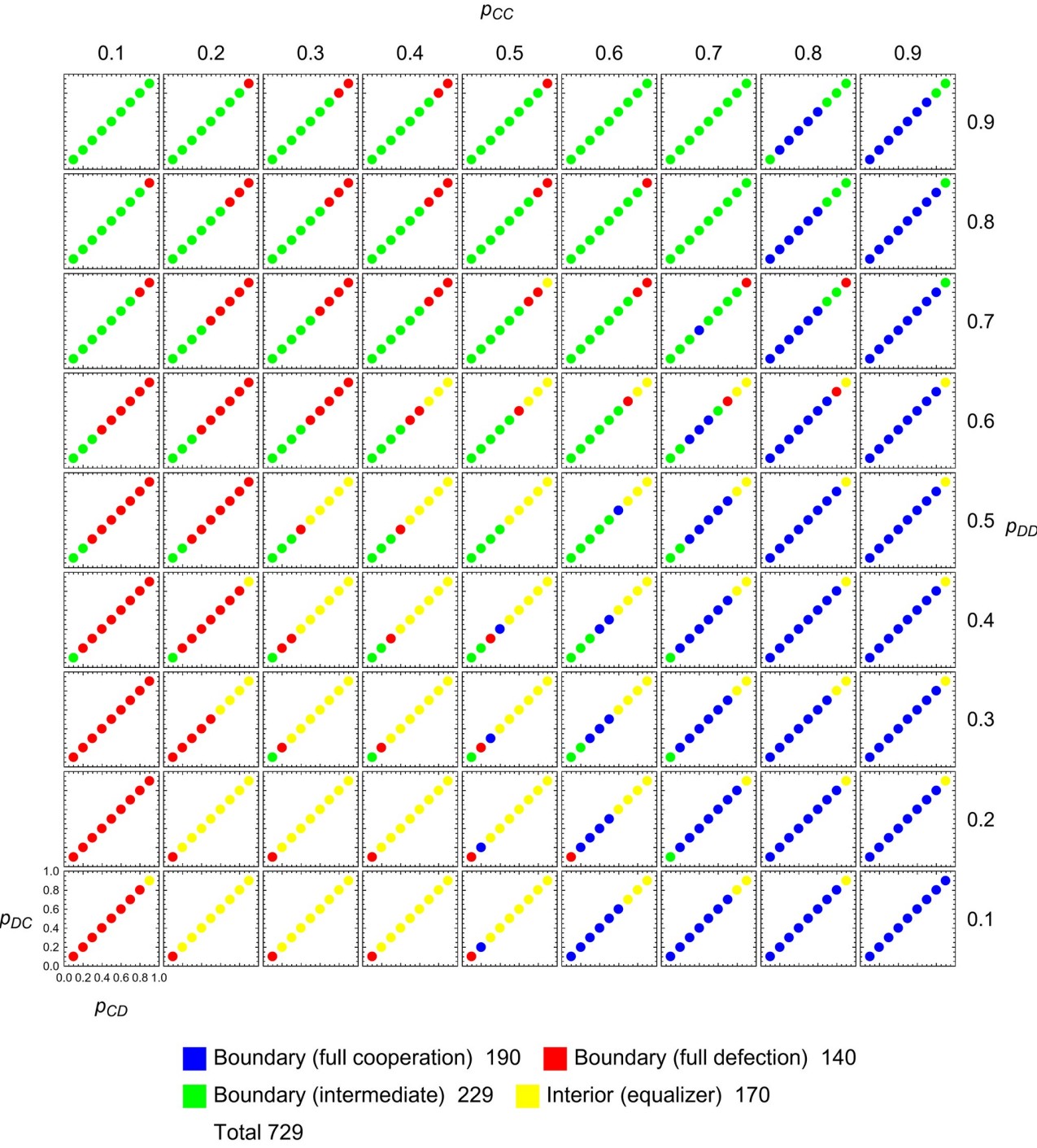

**Fig 7. Long-time limits of adaptive dynamics of counting strategies.** On a $9 \times 9 \times 9 \times 9$-grid representing the space of memory-one strategies, we depict the 729 points which are counting strategies (defined by $p_{CD} = p_{DC}$). They are colored according to the limit $\lim_{t \to \infty} q(t)$ of a solution $\mathbf{q}(t)$ to Eq (6), with starting value $\mathbf{q}(0)$ in the grid. Dynamics are assumed to cease at the boundary of the strategy space. Generically, there are 4 possibilities as shown in the legend. For 190 points the trajectory $\mathbf{q}(t)$ evolves to full cooperation, defined by $q_2 = 1$ (blue). For 140 points the trajectory $\mathbf{q}(t)$ evolves to full defection, defined by $q_0 = 0$ (red). The remaining points either evolve into other regions of the boundary (green) or approach interior critical points, which are equalizers (yellow). This figure is not a simple restriction of Fig 3 because the restriction of Eq (6) differs from Eq (12) by a factor of 2. Parameter $c = 0.1$.

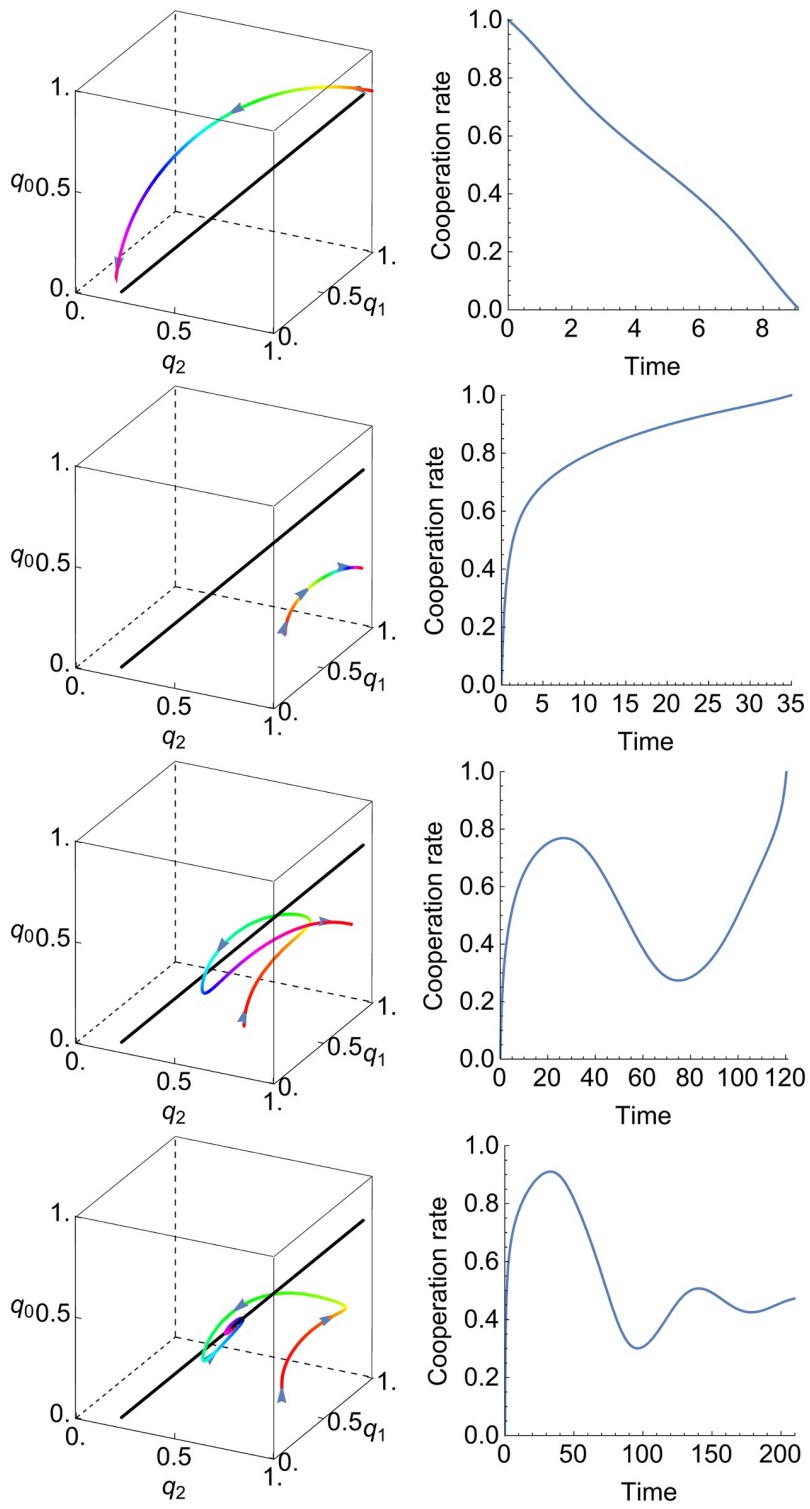

**Fig 8. Trajectories of adaptive dynamics of counting strategies.** We consider four different initial conditions. We plot the solutions $\mathbf{q}(t)$ to Eq (12) on the left, colored by hue and marked with arrowheads to indicate the direction of evolution in the strategy space. On the right, we plot the cooperation rate $C(\mathbf{q}(t))$, which is a real number between zero (full defection) and one (full cooperation). Each of the initial conditions leads to a different behavior. In the first row, for an initial condition $\mathbf{q}(0) = (1, 1, 0.8)$, the cooperation rate decreases monotonically from one to zero. In the second row, for $\mathbf{q}(0) = (0.6833, 0.85, 0)$, the cooperation rate increases monotonically from zero to one. In the third row, for $\mathbf{q}(0) = (0.6, 0.5, 0)$, the cooperation rate increases from zero to an intermediate value before decreasing and then

increasing again to one. Finally, in the last row, for $\mathbf{q}(0) = (0.6667, 0.75, 0)$, the cooperation rate increases from zero before oscillating and converging to an intermediate value. The last two orbits loop around the line of interior critical points, shown in black. Parameter $c = 0.1$.

We can also plot a few solutions $\mathbf{q}(t)$ of Eq (12) in three dimensions to give an idea of the possible behaviors. Four types of behavior are shown in Fig 8. Alongside plots of the trajectory $\mathbf{q}(t)$ we depict the cooperation rate $C(\mathbf{q}(t))$, defined as the average rate of cooperation in a large population playing the respective strategy. Previous studies show that these cooperation rates change monotonically when players are restricted to use reactive strategies (those with $p_{CC} = p_{DC}$ and $p_{CD} = p_{DD}$, see [1]). Within the counting strategies, this monotonicity is violated in the third and fourth example, and the fourth converges to intermediate cooperation rather than full cooperation or full defection.

## Discussion and conclusion

The donation game is one of the main paradigms to explore direct reciprocity, and memory-one strategies are among the best-studied strategy spaces in the respective literature [24–32]. These strategies are comparably simple. They only condition on the outcome of the very last round, while ignoring the outcome of all previous rounds.

Despite their simplicity, the formulas that describe the payoffs of memory-one players are non-trivial to manipulate mathematically. As a result, many previous studies on memory-one strategies rely on simulations. On the one hand, such simulations give valuable insights into the dynamics of reciprocity. On the other hand, they make it difficult to describe why certain strategies are favored by evolution, and how results depend on parameters such as the cost of cooperation.

To get a more analytical description of the evolution of reciprocity, we use the framework of adaptive dynamics. This framework considers homogeneous populations that move into the direction of mutants with maximum invasion fitness [52, 53]. For our setup of memory-one players in the donation game, we show that this dynamics has two remarkable mathematical properties. Our first result concerns the subspace of counting strategies. Counting strategies only depend on the number of cooperating players in the previous round. We show that the adaptive dynamics leaves the subspace of counting strategies invariant. Moreover, we show in Methods that this invariance result is not restricted to donation games or memory-one strategies. A similar invariance arises for arbitrary repeated $2 \times 2$ games, or when players remember more than the very last round.

Second, we describe an interesting symmetry between forward-in-time orbits and backward-in-time orbits. This symmetry is specific to the donation game, but is not restricted to memory-one strategies. Its importance becomes apparent in many of our figures (for example, in Figs 1 and 2, where it leads to beautiful geometric patterns).

We use these mathematical insights to qualitatively describe the adaptive dynamics of memory-one strategies and of counting strategies. In particular, we describe the set of interior critical points, and the set of saturated boundary points. Any converging solution of adaptive dynamics ends up in one of these two sets. While previous research has identified which memory-one strategies are Nash equilibria [38, 39], our study identifies those memory-one strategies that satisfy a local notion of uninvadability. For example, Eq (10) describes all memory-one strategies that are mutually cooperative and locally stable. The respective condition is less stringent than the condition for being a Nash equilibrium. This insight allows for the following interpretation. If evolution generates mutant strategies that are phenotypically similar to the

parent, there is a strictly larger strategy set of memory-one strategies that can maintain cooperation.

We believe these results give a more rigorous understanding of the properties of memory-one strategies. At the same time we hope that similar techniques can be used to explore other games and more general strategy spaces.

## Methods

### Adaptive dynamics of memory-one strategies

**Derivation of the adaptive dynamics.** In the main text, we have described how to define the payoff of two players with memory-one strategies by representing the game as a Markov chain. However, to derive the adaptive dynamics, it is useful to start with an alternative representation of the payoffs. As shown by [56], the payoff expression (5) can be rewritten as

$$A(\mathbf{p}, \mathbf{p}') = \frac{\det \begin{pmatrix} -1+p_{CC}p'_{CC} & -1+p_{CC} & -1+p'_{CC} & R \\ p_{CD}p'_{DC} & -1+p_{CD} & p'_{DC} & S \\ p_{DC}p'_{CD} & p_{DC} & -1+p'_{CD} & T \\ p_{DD}p'_{DD} & p_{DD} & p'_{DD} & P \end{pmatrix}}{\det \begin{pmatrix} -1+p_{CC}p'_{CC} & -1+p_{CC} & -1+p'_{CC} & 1 \\ p_{CD}p'_{DC} & -1+p_{CD} & p'_{DC} & 1 \\ p_{DC}p'_{CD} & p_{DC} & -1+p'_{CD} & 1 \\ p_{DD}p'_{DD} & p_{DD} & p'_{DD} & 1 \end{pmatrix}} \tag{15}$$

Using this representation, we can write out the expression for adaptive dynamics (6) in full. To this end, it is convenient to multiply the resulting system by the common denominator, $(1 - p_{CD} + p_{DC})r(p_{CC}, p_{CD}, p_{DC}, p_{DD})^2$, where

$$r(x, y, z, w) = w^2(-1 + 2x - y - z) + w(2 - 2x^2 + 2yz) + (-1 + x)(-1 + y + z - 2yz + x(-1 + y + z)) \tag{16}$$

This denominator is positive in the interior $(0, 1)^4$ of the strategy space. Hence, multiplying by the denominator only affects the timescale of evolution, but not the direction of the trajectories. After applying this modification to the system (6), the dynamics among the memory-one strategies of the donation game takes the following form,

$$\begin{aligned}
\dot{p}_{CC} &= f_1(p_{CD}, p_{DC}, p_{DD}) & \cdot & \left[ b \cdot g_1(p_{CC}, p_{CD}, p_{DC}, p_{DD}) + c \cdot h_1(p_{CC}, p_{CD}, p_{DC}, p_{DD}) \right] \\
\dot{p}_{CD} &= f_2(p_{CC}, p_{DD}) & \cdot & \left[ b \cdot g_2(p_{CC}, p_{CD}, p_{DC}, p_{DD}) + c \cdot h_2(p_{CC}, p_{CD}, p_{DC}, p_{DD}) \right] \\
\dot{p}_{DC} &= f_3(p_{CC}, p_{DD}) & \cdot & \left[ b \cdot g_3(p_{CC}, p_{CD}, p_{DC}, p_{DD}) + c \cdot h_3(p_{CC}, p_{CD}, p_{DC}, p_{DD}) \right] \\
\dot{p}_{DD} &= f_4(p_{CC}, p_{CD}, p_{DC}) & \cdot & \left[ b \cdot g_4(p_{CC}, p_{CD}, p_{DC}, p_{DD}) + c \cdot h_4(p_{CC}, p_{CD}, p_{DC}, p_{DD}) \right]
\end{aligned} \tag{17}$$

Here, the auxiliary functions $f_i, g_i, h_i$ for $i \in \{1, 2, 3, 4\}$ are defined as follows

$$
\begin{aligned}
f_1(y, z, w) &= w(2yz + w - yw - zw) \\
g_1(x, y, z, w) &= -w + wx - w^2x + x^2 + wx^2 + w^2y - xy - wxy - x^2y + xy^2 + z \\
&\quad -xz + wxz - wyz + xyz - y^2z - xz^2 + yz^2 \\
h_1(x, y, z, w) &= -1 - w + w^2 - wx - w^2x + wx^2 + y + xy + wxy - xy^2 + w^2z \\
&\quad +xz - wxz - x^2z - 2yz - wyz + xyz + y^2z + xz^2 - yz^2 \\
f_2(x, w) &= -w(1 - w + x)(1 - x) \\
g_2(x, y, z, w) &= w + w^2x - wx^2 - w^2y - z - wz - xz + x^2z + yz + 2wyz + z^2 \\
&\quad -wz^2 - xz^2 - yz^2 + z^3 \\
h_2(x, y, z, w) &= 1 + w - w^2 + w^2x - x^2 - wx^2 - y + x^2y + wz - w^2z + xz + yz \\
&\quad -2xyz - z^2 + wz^2 + xz^2 + yz^2 - z^3 \\
f_3(x, w) &= f_2(1 - w, 1 - x) \\
g_3(x, y, z, w) &= g_2(1 - w, 1 - z, 1 - y, 1 - x) \\
h_3(x, y, z, w) &= h_2(1 - w, 1 - z, 1 - y, 1 - x) \\
f_4(x, y, z) &= f_1(1 - z, 1 - y, 1 - x) \\
g_4(x, y, z, w) &= g_1(1 - w, 1 - y, 1 - z, 1 - x) \\
h_4(x, y, z, w) &= h_1(1 - w, 1 - z, 1 - y, 1 - x)
\end{aligned}
\tag{18}
$$

Note that we can write $f_i, g_i, h_i$ for $i \in \{3, 4\}$ in terms of the same functions for $i \in \{1, 2\}$. This is a consequence of the symmetry we discuss later.

**Invariance of counting strategies.** Using the representation (17) and (18), it becomes straightforward to show that the space of memory-one counting strategies remains invariant under adaptive dynamics.

**Proposition 1**. *Let $\mathcal{C}$ denote the three-dimensional subspace of counting strategies among the memory-one strategies,*

$$
\mathcal{C} := \{ \mathbf{p} \in [0, 1]^4 \mid p_{CD} = p_{DC} \}
\tag{19}
$$

*Then $\mathcal{C}$ is invariant under adaptive dynamics. That is, if $\mathbf{p}(t)$ is a solution of Eq (17) with $\mathbf{p}(0) \in \mathcal{C}$, then $\mathbf{p}(t) \in \mathcal{C}$ for all t.*

*Proof.* By using the definitions in (18), one can verify that

$$f_2(p_{CC}, p_{DD}) - f_3(p_{CC}, p_{DD}) = 0$$

$$g_2(p_{CC}, p_{CD}, p_{DC}, p_{DD}) - g_3(p_{CC}, p_{CD}, p_{DC}, p_{DD}) =$$

$$(1 - p_{CD} + p_{DC}) \quad (p_{CD} - p_{DC})(p_{CC} - p_{CD} - p_{DC} + p_{DD}) \quad (20)$$

$$h_2(p_{CC}, p_{CD}, p_{DC}, p_{DD}) - h_3(p_{CC}, p_{CD}, p_{DC}, p_{DD}) =$$

$$-(1 - p_{CD} + p_{DC}) \quad (p_{CD} - p_{DC})(p_{CC} - p_{CD} - p_{DC} + p_{DD})$$

In particular, if we define $d := p_{CD} - p_{DC}$, it follows by (17) and (20) that

$$\dot{d} = \dot{p}_{CD} - \dot{p}_{DC} = f_2(p_{CC}, p_{DD})(b - c)(1 - p_{CD} + p_{DC})(p_{CD} - p_{DC})(p_{CC} - p_{CD} - p_{DC} + p_{DD}) \quad (21)$$

For $d = p_{CD} - p_{DC} = 0$, we can therefore conclude that $\dot{d} = 0$.

While the proof of Proposition 1 shows that the set of counting strategies is invariant, it also shows that this set is not a local attractor. Instead, from Eq (21) it follows that the distance $d$ to the set of counting strategies decreases at a given time if and only if $\mathbf{p} \in (0, 1)^4$ satisfies $p_{CC} + p_{DD} > p_{CD} + p_{DC}$.

**A symmetry between forward and backward orbits.** Another direct implication of the functional form of adaptive dynamics in Eqs (17) and (18) is that solutions come in pairs. In Results we gave an intuitive argument for a symmetry in solutions for donation games. Here we derive the result formally.

**Proposition 2**. *Let $\mathbf{p}(t) = (p_{CC}, p_{CD}, p_{DC}, p_{DD})(t)$ be a solution to Eq (17) on some interval $t \in (a, b)$.*

*Then $\tilde{\mathbf{p}}(-t) := (1 - p_{DD}, 1 - p_{DC}, 1 - p_{CD}, 1 - p_{CC})(-t)$ is a solution to Eq (17) for the interval $t \in (-b, -a)$.*

*Proof.* We show the result for the first component; the other components follow similarly. For the first component, we have

$$
\begin{aligned}
\dot{\tilde{p}}_{CC}(-t) &= f_4(1 - p_{DD}, 1 - p_{DC}, 1 - p_{CD})[b\, g_4(1 - p_{DD}, 1 - p_{DC}, 1 - p_{CD}, 1 - p_{CC}) \\
&\quad + c\, h_4(1 - p_{DD}, 1 - p_{CD}, 1 - p_{DC}, 1 - p_{CC})] \\
&= f_1(p_{CD}, p_{DC}, p_{DD})\left[b \cdot g_1(p_{CC}, p_{CD}, p_{DC}, p_{DD}) + c \cdot h_1(p_{CC}, p_{CD}, p_{DC}, p_{DD})\right] \\
&= \dot{p}_{CC}(t)
\end{aligned}
$$

Therefore, if $\mathbf{p}(t)$ satisfies the differential Eq (17), then so does $\tilde{\mathbf{p}}(-t)$.

The transformation $\mathbf{p} \mapsto \tilde{\mathbf{p}}$, defined by $(p_{CC}, p_{CD}, p_{DC}, p_{DD}) \mapsto (1 - p_{DD}, 1 - p_{DC}, 1 - p_{CD}, 1 - p_{CC})$, reflects a point in the hypercube $[0, 1]^4$ with respect to the 2-dimensional plane

$$\mathcal{P} = \left\{ \mathbf{p} \in [0, 1]^4 \ \middle|\ p_{CC} + p_{DD} = 1, \ p_{CD} + p_{DC} = 1 \right\} \quad (22)$$

That is, if one takes the line segment between $\mathbf{p}$ and $\tilde{\mathbf{p}}$, then the midpoint of this line segment is in $\mathcal{P}$. The plane $\mathcal{P}$ is exactly the set of points that are mapped onto themselves. Every point is mapped onto itself if the transformation is applied twice. It can be directly checked that the transformation $\mathbf{p} \mapsto \tilde{\mathbf{p}}$ maps critical points to critical points (see next subsection), and the previous proposition means that it interchanges points which are limits forward in time and points which are limits backward in time.

The symmetry described by Proposition 2 is not unique to memory-one strategies; it is a general phenomenon related to equal gains from switching. For example, the same argument we used in Results can be used to establish a direct analogue of Proposition 2 for memory-one counting strategies and for memory-$n$ strategies.

The symmetry is particularly easy to visualize for the three-dimensional space of memory-one counting strategies. In this case, we define $\mathbf{q} \to \tilde{\mathbf{q}}$ to be the transformation $(q_2, q_1, q_0) \mapsto (1-q_0, 1-q_1, 1-q_2)$. The analogue of Proposition 2 says that if $\mathbf{q}(t)$ is a solution to 12 on the interval $t \in (a, b)$, then so is $\tilde{\mathbf{q}}(-t)$ on the interval $t \in (-b, -a)$. This pair of solutions is related by a time reversal and a rotation of the cube $[0, 1]^3$ about the axis $q_1 = 1/2$, $q_2 + q_0 = 1$.

**Critical points of adaptive dynamics.**   In the following, we characterize the fixed (critical) points of adaptive dynamics in the interior of the hypercube.

**Proposition 3**. *A stochastic strategy* $\mathbf{p} \in (0, 1)^4$ *is a critical point of system* (17) *if and only if*

$$b(p_{CC} - p_{CD}) - c(1 - p_{CC} + p_{DC}) = 0, \quad p_{CC} + p_{DD} = p_{CD} + p_{DC} \tag{23}$$

*Proof.* ($\Rightarrow$) Directly setting $0 = \dot{p}_{CC} = \dot{p}_{CD} = \dot{p}_{DC} = \dot{p}_{DD}$ quickly becomes unwieldy. Notice, however, that $f_1, f_2, f_3, f_4$ do not vanish when their parameters take values in $(0, 1)$. So at interior critical points, we must have

$$
\begin{aligned}
0 &= \frac{\dot{p}_{CC}}{f_1(p_{CD}, p_{DC}, p_{DD})} + \frac{\dot{p}_{CD}}{f_2(p_{CC}, p_{DD})} \\
&= (b - c)(p_{CC} - p_{DC})(p_{CC} + p_{DD} - p_{CD} - p_{DC})(1 - p_{CD} + p_{DC}) \\
0 &= \frac{\dot{p}_{CC}}{f_1(p_{CD}, p_{DC}, p_{DD})} + \frac{\dot{p}_{DC}}{f_3(p_{CC}, p_{DD})} \\
&= (b - c)(p_{CC} - p_{CD})(p_{CC} + p_{DD} - p_{CD} - p_{DC})(1 - p_{CD} + p_{DC}) \\
0 &= \frac{\dot{p}_{CC}}{f_1(p_{CD}, p_{DC}, p_{DD})} - \frac{\dot{p}_{DD}}{f_4(p_{CC}, p_{CD}, p_{DC})} \\
&= (b - c)(p_{CC} - p_{DD})(p_{CC} + p_{DD} - p_{CD} - p_{DC})(1 - p_{CD} + p_{DC})
\end{aligned}
\tag{24}
$$

Since $1 - p_{CD} + p_{DC} > 0$ for $p_{CD}, p_{DC} \in (0, 1)$, either $p_{CC} = p_{CD} = p_{DC} = p_{DD}$ or $p_{CC} + p_{DD} = p_{CD} + p_{DC}$ must be enforced. Note that if $p_{CC} = p_{CD} = p_{DC} = p_{DD}$, then $p_{CC} + p_{DD} = p_{CD} + p_{DC}$ holds trivially. Hence, in both cases we have the identity $p_{DD} = p_{CD} + p_{DC} - p_{CC}$, which we can plug into $\dot{p}_{CC}/f_1(p_{CD}, p_{DC}, p_{DD})$ to get

$$\frac{\dot{p}_{CC}}{f_1(p_{CD}, p_{DC}, p_{DD})} = \left(b(p_{CD} - p_{CC}) + c(1 - p_{CC} + p_{DC})\right) \cdot \left(-1 + (p_{CD} - p_{CC})^2 + (p_{DC} - p_{CC})^2\right) \tag{25}$$

It is verified without too much difficulty that whenever the second factor vanishes in $(0, 1)^3$, then $p_{CD} + p_{DC} - p_{CC} \notin (0, 1)$. Any interior critical points of (17) thus needs to satisfy

$$b(p_{CC} - p_{CD}) - c(1 - p_{CC} + p_{DC}) = 0 \quad \text{and} \quad p_{CC} + p_{DD} = p_{CD} + p_{DC} \tag{26}$$

($\Leftarrow$) If a strategy satisfies the conditions (26), we can express $p_{CD}$ and $p_{DC}$ in terms of $p_{CC}$ and $p_{DD}$,

$$p_{CD} = \frac{b\, p_{CC} - c(1 + p_{DD})}{b - c} \quad \text{and} \quad p_{DC} = \frac{c(1 - p_{CC}) + b p_{DD}}{b - c} \tag{27}$$

Inserting these expressions into the system (17) yields, after some algebraic manipulations, $\dot{p}_{CC} = \dot{p}_{CD} = \dot{p}_{DC} = \dot{p}_{DD} = 0$.

Solving Eq (23) for $p_{CC}$ and $p_{DD}$, we arrive at

$$p_{CC} = \frac{c + b\, p_{CD} + c\, p_{DC}}{b + c} \quad \text{and} \quad p_{DD} = \frac{c\,(-1 + p_{CD}) + b\, p_{DC}}{b + c} \tag{28}$$

Using (28), the constraint $p_{DD} > 0$ becomes $p_{DC} > (c/b)(1 - p_{CD})$. When we plug this back into the expression for $p_{CC}$ and use the fact that $p_{CD} > 0$, we get $p_{CC} > c/b$. Similarly, the constraints $p_{CC} < 1$ and $p_{DC} < 1$ lead to $p_{DD} < 1 - c/b$. The result is that we have two useful bounds $p_{CC} > c/b$ and $p_{DD} < 1 - c/b$ among the interior critical points.

We now relate the interior critical points to the equalizer strategies discussed by [57] and [56].

**Definition**. *An equalizer is a strategy $\mathbf{p}$ for which $A(\mathbf{p}', \mathbf{p})$ is a constant function of $\mathbf{p}'$.*

It follows from the definition that every equalizer strategy is a critical point of the dynamics (17). In the interior $(0, 1)^4$, the converse is also true. That is,

**Proposition 4**. *Every interior critical point of the system (17) is an equalizer.*

*Proof*. Our condition for critical points (27) coincides with the expression for equalizers, Eq. (8) in [56], when using the payoffs of the donation game.

As shown by [39], equalizers are the only Nash equilibria among the stochastic memory-one strategies. Thus our above results can be summarized as follows. In the donation game, an interior point is a critical point of adaptive dynamics if and only if it is a Nash equilibrium (such a result does not need to hold in general, because strategies might be locally stable critical points of adaptive dynamics without being global best responses to themselves, see [50]).

**Analysis of the boundary faces.** In the main text, we define the boundary of the strategy space $[0, 1]^4$ as the set of all $(p_{CC}, p_{CD}, p_{DC}, p_{DD})$ for which exactly one entry is in $\{0, 1\}$. Therefore there are eight different boundary faces. One particularly important face is the one with $p_{CC} = 1$, which corresponds to a fully cooperative population. It follows from Eq (18) that on this boundary face $f_2(p_{CC}, p_{DD}) = f_3(p_{CC}, p_{DD}) = f_4(p_{CC}, p_{CD}, p_{DC}) = 0$. By Eq (17) we can then conclude that $\dot{p}_{CD} = \dot{p}_{DC} = \dot{p}_{DD} = 0$. A point $\mathbf{p}$ on this boundary face is saturated if and only if $\dot{p}_{CC} \geq 0$. By Eq (17) and because $f_1(p_{CD}, p_{DC}, p_{DD}) > 0$, this condition is equivalent to $b \cdot g_1(1, p_{CD}, p_{DC}, p_{DD}) > -c \cdot h_1(1, p_{CD}, p_{DC}, p_{DD})$, which yields condition (10).

The boundary face with $p_{DD} = 0$ can be analyzed analogously.

## Adaptive dynamics of memory-one counting strategies

In the following, we identify memory-one counting strategies with points in the 3-dimensional cube $[0, 1]^3$. The entries of a counting strategy $\mathbf{q} = (q_2, q_1, q_0)$ correspond to the cooperation probability in the next round, based on the number of cooperators in the previous round. We can embed the space of counting strategies into the space of memory-one strategies by using the mapping $(q_2, q_1, q_0) \mapsto (q_2, q_1, q_1, q_0)$. Using this embedding, we can compute the payoff of

a $\mathbf{q}$-player against $\mathbf{q}'$-player using the payoff formula (15), which yields

$$A(\mathbf{q}, \mathbf{q}') = \frac{\det \begin{pmatrix} -1 + q_2 q_2' & -1 + q_2 & -1 + q_2' & b - c \\ q_1 q_1' & -1 + q_1 & q_1' & -c \\ q_1 q_1' & q_1 & -1 + q_1' & b \\ q_0 q_0' & q_0 & q_0' & 0 \end{pmatrix}}{\det \begin{pmatrix} -1 + q_2 q_2' & -1 + q_2 & -1 + q_2' & 1 \\ q_1 q_1' & -1 + q_1 & q_1' & 1 \\ q_1 q_1' & q_1 & -1 + q_1' & 1 \\ q_0 q_0' & q_0 & q_0' & 1 \end{pmatrix}} \tag{29}$$

In the following we study the adaptive dynamics of counting strategies. Again, we consider a homogeneous population with strategy $\mathbf{q}$, evolving in the direction of the gradient of the payoff function, now calculated in $[0, 1]^3$. Evolution in the space of counting strategies is thus given by

$$\dot{q}_i = \frac{\partial A(\mathbf{q}, \mathbf{q}')}{\partial q_i}\bigg|_{\mathbf{q}=\mathbf{q}'} \tag{30}$$

To write out the adaptive dynamics Eq (30) in full, it is again convenient to multiply the equations by the common denominator $r(q_2, q_1, q_0)^2$, with

$$r(x, y, z) = (-1+x)(-1+y+(1-2y)(y-x)) + (2 - 2x^2 + 2y^2)z + (-1 + 2x - 2y)z^2 \tag{31}$$

This denominator is nonzero in the interior $(0, 1)^3$ of the strategy space. After this rescaling, the system of Eq (30) becomes

$$\dot{q}_2 = f_2(q_1, q_0) \cdot \left[ b \cdot g_2(q_2, q_1, q_0) + c \cdot h_2(q_2, q_1, q_0) \right]$$

$$\dot{q}_1 = f_1(q_2, q_0) \cdot \left[ b \cdot g_1(q_2, q_1, q_0) + c \cdot h_1(q_2, q_1, q_0) \right] \tag{32}$$

$$\dot{q}_0 = f_0(q_2, q_1) \cdot \left[ b \cdot g_0(q_2, q_1, q_0) + c \cdot h_0(q_2, q_1, q_0) \right]$$

The auxiliary functions $f_i, g_i, h_i$ now take the form

$$f_2(y, z) = -z(2y(y - z) + z)$$

$$g_2(x, y, z) = -x^2 - y + 2xy + x^2 y - xy^2 + z - xz - x^2 z + y^2 z + xz^2 - yz^2$$

$$h_2(x, y, z) = 1 - y - 2xy + x^2 y + 2y^2 - xy^2 + z + xz - x^2 z + y^2 z - z^2 + xz^2 - yz^2$$

$$f_1(x, z) = -2z(-1 + x)(1 + x - z)$$

$$g_1(x, y, z) = y + xy - x^2 y - 2y^2 + xy^2 - z + x^2 z + yz - y^2 z - xz^2 + yz^2 \tag{33}$$

$$h_1(x, y, z) = -1 + x^2 + y - xy - x^2 y + xy^2 - z + x^2 z - yz - y^2 z + z^2 - xz^2 + yz^2$$

$$f_0(x, y) = f_2(1 - y, 1 - x)$$

$$g_0(x, y, z) = g_2(1 - z, 1 - y, 1 - x)$$

$$h_0(x, y, z) = h_2(1 - z, 1 - y, 1 - x)$$

**Critical points of adaptive dynamics of counting strategies.** Again, in the following we characterize the fixed (critical) points of adaptive dynamics in the interior of $[0, 1]^3$.

**Proposition 5**. *The interior critical points of the system* (32) *are parametrized by*

$$\left(t + \frac{c}{b+c}, t, t - \frac{c}{b+c}\right), \quad for \ \ t \in \left(\frac{c}{b+c}, \ \frac{b}{b+c}\right) \tag{34}$$

*Proof.* Because $f_2, f_1, f_0$ do not vanish in the interior of the strategy space $(0, 1)^3$, we can compute

$$\frac{\dot{q}_1}{f_1(q_2, q_0)} + \frac{\dot{q}_0}{f_0(q_2, q_1)} = (b - c)(q_0 - q_1)(q_2 - 2q_1 + q_0),$$

$$\frac{\dot{q}_2}{f_2(q_1, q_0)} - \frac{\dot{q}_0}{f_0(q_2, q_1)} = (b - c)(q_2 - q_0)(q_2 - 2q_1 + q_0) \tag{35}$$

At a critical point we have $\dot{q}_2 = \dot{q}_1 = \dot{q}_0 = 0$, so the expressions on the right hand side must vanish. This implies $q_2 - 2q_1 + q_0 = 0$ or $q_2 = q_1 = q_0$ (in which case $q_2 - 2q_1 + q_0 = 0$ holds trivially). So $q_1 = (q_2 + q_0)/2$ is a necessary condition for the strategy **q** to be a critical point. To obtain a condition that is also sufficient we take this expression for $q_1$ and plug it into

$$\frac{4\dot{q}_1(q_2, (q_2+q_0)/2, q_0)}{f_1(q_2, q_0)} = \left(b(q_0 - q_2) + c(2 + q_0 - q_2)\right)\left(2 - (q_2 - q_0)^2\right) \tag{36}$$

This expression only vanishes when $q_2 - q_0 = \frac{2c}{b+c}$. The solutions to the conditions

$$q_2 + q_0 = 2q_1, \quad q_2 - q_0 = \frac{2c}{b+c} \tag{37}$$

are parameterized by

$$\left(t + \frac{c}{b+c}, t, t - \frac{c}{b+c}\right), \ \ t \in \left(\frac{c}{b+c}, \frac{b}{b+c}\right) \tag{38}$$

Conversely, it is easily checked that all of these strategies are critical points of (32).

Thus the interior critical points form a straight line segment on the interior of the cube with boundary points $\left(\frac{2c}{b+c}, \frac{c}{b+c}, 0\right)$ and $\left(1, \frac{b}{b+c}, \frac{b-c}{b+c}\right)$ and length $\sqrt{3}\frac{b-c}{b+c}$, which ranges from $\sqrt{3}$ (the diagonal of the cube) to 0 as $\frac{c}{b}$ ranges from 0 to 1. We can classify the stability of these critical points by finding their associated eigenvalues. The results are complicated, but shown in Fig 5.

## Comparison to reactive strategies

Reactive strategies are the memory-one strategies satisfying $p_{CC} = p_{DC}$ and $p_{CD} = p_{DD}$. They form a two-dimensional space which has been studied extensively, including their adaptive dynamics [43–48]. The set of interior critical points for adaptive dynamics of reactive strategies coincides with the set of equalizer strategies, a result which we generalized in Results.

However, we also highlight several key differences between the strategy spaces. One important theme is that the three-dimensional space of memory-one counting strategies captures a surprising degree of complexity not seen in reactive strategies. In Fig 8 we show that the rate of self-cooperation does not always monotonically increase or decrease, as it does for reactive strategies. In fact, cooperativity can increase and decrease several times along a trajectory. Furthermore, the symmetry $\mathbf{p}(t) \mapsto \tilde{\mathbf{p}}(-t)$ has a direct analogue for reactive strategies, which

turns out to associate each trajectory to itself. That is, trajectories for reactive strategies do not come in pairs, as they do in the larger spaces of memory-one, memory-one counting, and higher memory strategies.

In Fig 9, we plot the cooperative region for memory-one strategies (the region for which the self-cooperation rate is locally increasing). The corresponding region for reactive strategies is straightforward to describe [43]: If $(p_C, p_D)$ is a player's probability to cooperate depending on the co-player's previous action ($C$ or $D$), then the cooperative region consists of all points with $p_C - p_D > c/b$.

## Extensions of the invariance result

Our Proposition 1 shows that among the memory-one strategies of the donation game, adaptive dynamics leaves the set of counting strategies invariant. In the following, we derive two generalizations of this result. In a first step, we show that the same result holds for arbitrary repeated $2 \times 2$ games.

**Proposition 6**. *Let $\mathcal{C}$ denote the three-dimensional subspace of counting strategies among the memory-one strategies, as defined by* Eq (19). *Then $\mathcal{C}$ is invariant under adaptive dynamics, for any repeated $2 \times 2$ game with payoff matrix* (2).

*Proof.* Let $M$ be the Markov chain of the form (4) generated by the behavior of two players with strategies $\mathbf{p}$ and $\mathbf{p}'$. Moreover, let $\mathbf{v}$ denote the associated stationary distribution. The payoff to the $\mathbf{p}$-player in the repeated $2 \times 2$ game is then given by $A(\mathbf{p}, \mathbf{p}') = \pi(\mathbf{v})$, where $\pi : \mathbb{R}^4 \to \mathbb{R}$ is some linear map that depends on the payoff matrix of the game but not on $\mathbf{p}$ or $\mathbf{p}'$.

By definition $\mathbf{v}M = \mathbf{v}$. If we introduce an infinitesimal variation $\delta \mathbf{p}$ in the strategy $\mathbf{p}$ there will be an associated $\delta M$ and $\delta \mathbf{v}$, and they satisfy $(\mathbf{v} + \delta\mathbf{v})(M + \delta M) = \mathbf{v} + \delta\mathbf{v}$. Since $\mathbf{v} M = \mathbf{v}$ and since $\delta\mathbf{v}\delta M$ is disregarded as doubly infinitesimal, we have $\delta \mathbf{v} M + \mathbf{v}\delta M = \delta\mathbf{v}$. Choose $\delta \mathbf{p}$ to be $(0, \epsilon, -\epsilon, 0)$. Then it can be seen easily that

$$\delta M = \begin{pmatrix} 0 & 0 & 0 & 0 \\ \epsilon p'_{DC} & \epsilon(1 - p'_{DC}) & -\epsilon p'_{DC} & -\epsilon(1 - p'_{DC}) \\ -\epsilon p'_{CD} & -\epsilon(1 - p'_{CD}) & \epsilon p'_{CD} & \epsilon(1 - p'_{CD}) \\ 0 & 0 & 0 & 0 \end{pmatrix} \tag{39}$$

Now suppose $\mathbf{p}$ and $\mathbf{p}'$ are equal and furthermore that $p_{CD} = p_{DC}$. Then $v_{CD} = v_{DC}$ by symmetry, and $\mathbf{v}\delta M$ manifestly vanishes. It follows from the above that $\delta\mathbf{v}M = \delta\mathbf{v}$. Then $\delta\mathbf{v}$ is proportional to $\mathbf{v}$ by uniqueness of a stationary distribution. But we are also demanding that the sum of components of $\mathbf{v} + \delta\mathbf{v}$ is 1. Thus $\delta\mathbf{v} = 0$ and there is no variation in payoff $\pi(\mathbf{v})$. No player gains from deviating infinitesimally off the hypersurface $p_{CD} = p_{DC}$ in adaptive dynamics, i.e. from departing the space $\mathcal{C}$.

In a second step, we ask whether a similar invariance result applies to memory-$n$ strategies. With an argument similar to the one above, we can show that it applies at least in a restricted way.

Our notation for memory-$n$ strategies is best introduced by example: the component $p\begin{pmatrix} CDC \\ DDC \end{pmatrix}$ of a memory-3 strategy of player 1 denotes the probability of cooperation if the outcomes of the most recent three rounds were $CD$, $DD$, $CC$, in that order.

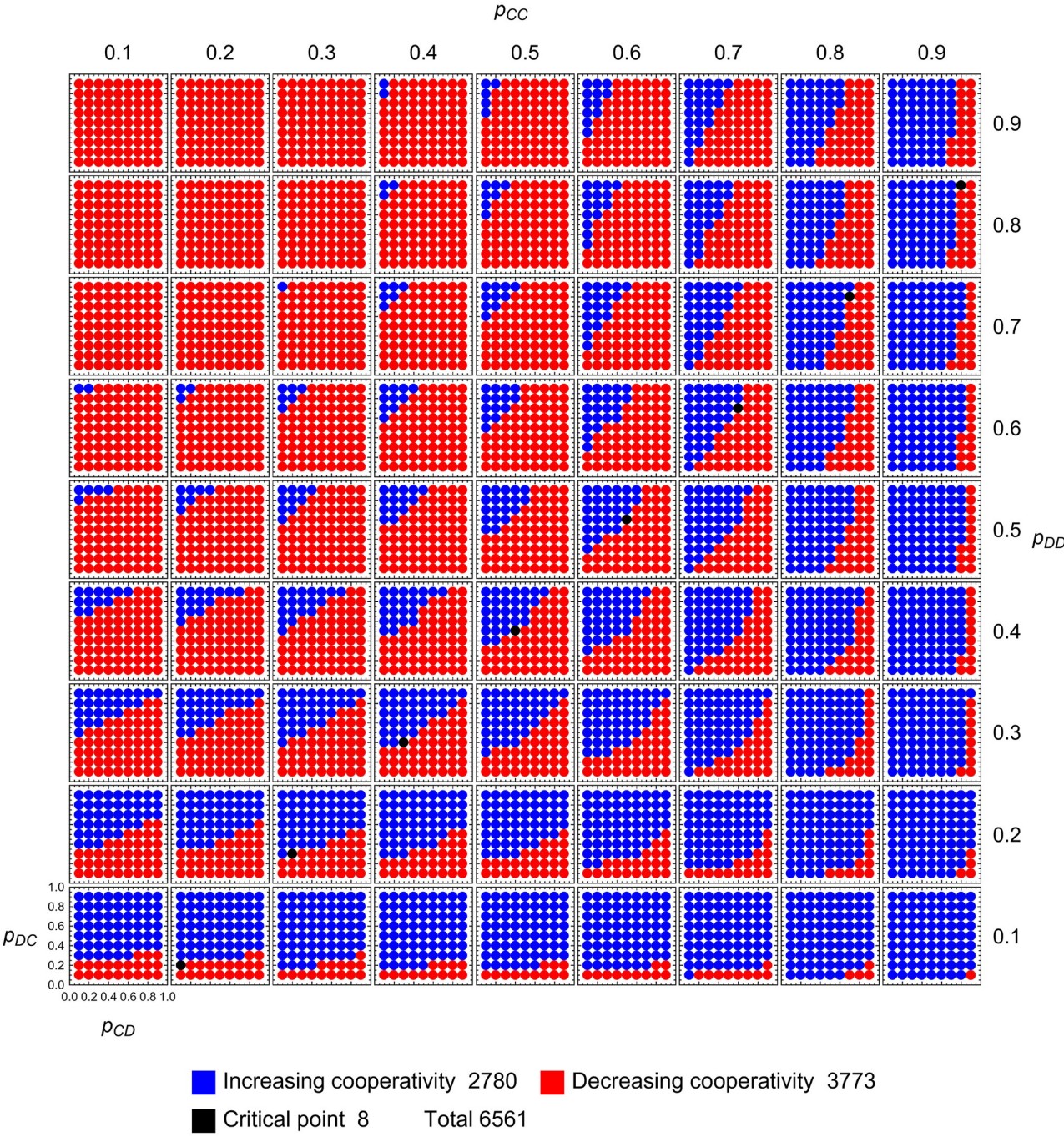

**Fig 9. Cooperative region for adaptive dynamics of memory-one strategies.** For a $9 \times 9 \times 9 \times 9$-grid (= 6561 points) we show the points for which the cooperativity, or rate of self-cooperation, of $(\dot{p}_{CC}, \dot{p}_{CD}, \dot{p}_{DC}, \dot{p}_{DD})$ is locally increasing. The rate of self-cooperation of a strategy **p** can be calculated by $A(\mathbf{p}, \mathbf{p})/(b - c)$ using formula (15). We find that for 1876 points cooperativity is locally increasing; for 4677 points cooperativity is decreasing; and eight points are critical points with $(\dot{p}_{CC}, \dot{p}_{CD}, \dot{p}_{DC}, \dot{p}_{DD}) = (0, 0, 0, 0)$. Note that, unlike the corresponding region for reactive strategies, trajectories beginning in the cooperative region can leave this region, and trajectories beginning outside of the cooperative region can enter it. We show examples of this in Fig 8). The graph is created for $c = 0.1$.

**Proposition 7**. *Consider the adaptive dynamics for memory-n strategies* $\mathbf{p}$ *and let s be a fixed arbitrary sequence of n − 1 moves for one player. Then the condition*

$$p\begin{pmatrix} Cs \\ Ds \end{pmatrix} = p\begin{pmatrix} Ds \\ Cs \end{pmatrix} \tag{40}$$

*is invariant for any repeated* $2 \times 2$ *game.*

*Proof.* Similar to before, let $M$ be the Markov chain generated by the behavior of two players with memory-$n$ strategies $\mathbf{p}$ and $\mathbf{p}'$, with stationary distribution $\mathbf{v}$. The components of $\mathbf{v}$ are the average frequencies of observing each possible history of length $n$ over the course of the game. The payoff to player 1 is given by $A(S, S') = \pi(\mathbf{v})$, where $\pi : \mathbb{R}^{4^n} \to \mathbb{R}$ is again some linear function depending on the payoff matrix of the game but independent of $\mathbf{p}$ and $\mathbf{p}'$. Again, we introduce an infinitesimal variation $\delta\mathbf{p}$ in the strategy $\mathbf{p}$. As a result, there will be an associated $\delta M$ and $\delta\mathbf{v}$, and they satisfy $(\mathbf{v} + \delta\mathbf{v})(M + \delta M) = \mathbf{v} + \delta\mathbf{v}$. Since $\mathbf{v}M = \mathbf{v}$, and $\delta\mathbf{v}\delta M$ is disregarded as doubly infinitesimal, we have $\delta\mathbf{v}M + \mathbf{v}\,\delta M = \delta\mathbf{v}$.

Now suppose that $\mathbf{p}$ is a memory-$n$ strategy that satisfies condition (40), with $s$ being an arbitrary but fixed sequence of length $n − 1$ of $C$'s and $D$'s. Let $\mathbf{e}_i$ denote the vector with a 1 in the $i$th position and zeros elsewhere, and let $\mathbf{e}_{i,j}$ denote the matrix with a 1 in the $i, j$'th entry and zeros elsewhere. The dimensions will be clear from context. We introduce the following infinitesimal variation in $\mathbf{p}$,

$$\delta\mathbf{p} = \epsilon \cdot \mathbf{e}\begin{pmatrix} Cs \\ Ds \end{pmatrix} - \epsilon \cdot \mathbf{e}\begin{pmatrix} Ds \\ Cs \end{pmatrix} \tag{41}$$

The corresponding variation in $M$ is

$$
\begin{aligned}
\delta M \;=\; & \epsilon p'\begin{pmatrix} Ds \\ Cs \end{pmatrix} \mathbf{e}\begin{pmatrix} Cs & sD \\ Ds & sC \end{pmatrix} - \epsilon\left(1 - p'\begin{pmatrix} Ds \\ Cs \end{pmatrix}\right)\mathbf{e}\begin{pmatrix} Cs & sC \\ Ds & sD \end{pmatrix} \\[2ex]
& - \epsilon p'\begin{pmatrix} Ds \\ Cs \end{pmatrix} \mathbf{e}\begin{pmatrix} Cs & sD \\ Ds & sC \end{pmatrix} - \epsilon\left(1 - p'\begin{pmatrix} Ds \\ Cs \end{pmatrix}\right)\mathbf{e}\begin{pmatrix} Cs & sD \\ Ds & sD \end{pmatrix} \\[2ex]
& - \epsilon p'\begin{pmatrix} Cs \\ Ds \end{pmatrix} \mathbf{e}\begin{pmatrix} Ds & sC \\ Cs & sC \end{pmatrix} - \epsilon\left(1 - p'\begin{pmatrix} Cs \\ Ds \end{pmatrix}\right)\mathbf{e}\begin{pmatrix} Ds & sC \\ Cs & sD \end{pmatrix} \\[2ex]
& + \epsilon p'\begin{pmatrix} Cs \\ Ds \end{pmatrix} \mathbf{e}\begin{pmatrix} Ds & sD \\ Cs & sC \end{pmatrix} + \epsilon\left(1 - p'\begin{pmatrix} Cs \\ Ds \end{pmatrix}\right)\mathbf{e}\begin{pmatrix} Ds & sD \\ Cs & sD \end{pmatrix}.
\end{aligned} \tag{42}
$$

We can compute

$$
\begin{aligned}
\mathbf{v}\delta M \quad =\; & \epsilon\left[ v\binom{Cs}{Ds}^{p'}\binom{Ds}{Cs} - v\binom{Ds}{Cs}^{p'}\binom{Cs}{Ds}\right]\mathbf{e}\binom{sC}{sC} \\[6pt]
& +\epsilon\left[ v\binom{Cs}{Ds}\left(1-p'\binom{Ds}{Cs}\right) - v\binom{Ds}{Cs}\left(1-p'\binom{Cs}{Ds}\right)\right]\mathbf{e}\binom{sC}{sD} \\[6pt]
& +\epsilon\left[ -v\binom{Cs}{Ds}^{p'}\binom{Ds}{Cs} + v\binom{Ds}{Cs}^{p'}\binom{Cs}{Ds}\right]\mathbf{e}\binom{sD}{sC} \\[6pt]
& +\epsilon\left[ -v\binom{Cs}{Ds}\left(1-p'\binom{Ds}{Cs}\right) + v\binom{Ds}{Cs}\left(1-p'\binom{Cs}{Ds}\right)\right]\mathbf{e}\binom{sD}{sD}
\end{aligned}
\tag{43}
$$

If $\mathbf{p}$ and $\mathbf{p}'$ are equal, then it follows by symmetry that

$$
v\binom{Cs}{Ds} = v\binom{Ds}{Cs}
\tag{44}
$$

Now (40) applied to $\mathbf{p}'$, along with (44), imply that the right hand side of (43) vanishes. Since $\mathbf{v}\delta M = 0$, our initial discussion means that $\delta\mathbf{v}M = \delta\mathbf{v}$. Therefore $\delta\mathbf{v}$ is proportional to $\mathbf{v}$ by uniqueness of stationary distribution. Because the sum of components of $\mathbf{v} + \delta\mathbf{v}$ is 1, we conclude that $\delta\mathbf{v} = 0$. Hence there is no variation in payoff $\pi(\mathbf{v})$. No player gains from making the infinitesimal variation (41).

## Author Contributions

**Conceptualization:** Philip LaPorte, Christian Hilbe, Martin A. Nowak.

**Formal analysis:** Philip LaPorte.

**Supervision:** Christian Hilbe, Martin A. Nowak.

**Validation:** Christian Hilbe, Martin A. Nowak.

**Visualization:** Philip LaPorte, Martin A. Nowak.

**Writing – original draft:** Philip LaPorte, Christian Hilbe, Martin A. Nowak.

**Writing – review & editing:** Philip LaPorte, Christian Hilbe, Martin A. Nowak.

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
