## [Decision Letter · Decision Letter 0]

16 Apr 2023

Dear Dr. Hilbe,

Thank you very much for submitting your manuscript "Adaptive dynamics of memory-1 strategies in the repeated donation game" for consideration at PLOS Computational Biology. As with all papers reviewed by the journal, your manuscript was reviewed by members of the editorial board and by several independent reviewers. The reviewers appreciated the attention to an important topic. Based on the reviews, we are likely to accept this manuscript for publication, providing that you modify the manuscript according to the review recommendations.

Sincerely,

Feng Fu

Academic Editor

PLOS Computational Biology

James O'Dwyer

Section Editor

PLOS Computational Biology

Reviewer's Responses to Questions

**Comments to the Authors:**

Reviewer #1: This work, taking a fair theoretical approach that seems scientifically healthy, successfully derives that 4D space of memory-1 strategies (denoted by (p_P, p_R, p_S, p_T) where one cooperates with the probability of p_{P,R,S,T} when he obtains {P,R,S,T} in the previous time-step) has an invariant 3D subspace (denoted by (q_0, q_1, q_2) where q_i is the probability to cooperate if i of the two players (focal agent and his opponent) have cooperated in the previous time-step).

Qualitatively speaking, 1-memory strategy allows a focal player to evaluate his strategy by the summed-up payoff over past two games, not just by the consequence of a single game. This enables to evolve the so-called ST-reciprocity where one of the two defects while his opponent cooperates (thus, he gains T while the opponent gains S), and subsequently (alternatively in time-direction), he cooperates while his opponent defects (thus, he gains S while the opponent gains T). This is more efficient than R-reciprocity (where both players repeat to cooperate, which brings R to them) if S + T > 2R.

Yet, a donation game (borrowing the authors terminology); Donner & Recipient (D & R) game (which many biologists have favored to call) in other words, where S + T = 2R because of R := b – c, T := b, P := 0, and S := - c) is neutral for whether ST-reciprocity or R-reciprocity being favored.

This MS can be published with the current content, since the authors’ methodology supported by several visual results seems solid and reliable. Yet, I would like to suggest the authors to mention about R-reciprocity and ST-reciprocity abovementioned (perhaps with some citations).

Reviewer #2: The iterated Prisoner’s Dilemma, along with the corresponding strategies, is a rich and dynamic subject of study and continues to generate interest and new research in various fields, including both evolutionary biology and computer science. The development of sophisticated strategies and the study of their effectiveness in different contexts are also ongoing areas of investigation. Even with the vast amount of research dedicated to the iterated Prisoner’s Dilemma and memory-one strategies (which are probably the simplest among all strategies), there are still significant gaps in our understanding of their behavior.

In this paper, the authors discuss the adaptive dynamics of memory-one strategies in the repeated donation game (a particular type of Prisoner’s Dilemma satisfying ‘equal gains from switching’) from an analytical perspective. In particular, they find two interesting mathematical results. First, they show that the four-dimensional space of memory-one strategies contains an invariant three-dimensional subset under adaptive dynamics. The subset is referred to as the collection of ‘counting strategies’, corresponding to memory-one strategies with p_CD = p_DC. Second, they prove that the adaptive dynamics exhibits a symmetry between orbits froward-in-time and backward-in-time for the donation game. There findings are based on strict mathematical expressions which can be used to characterize the adaptive dynamics among memory-one strategies.

The research presented in this paper represents a noteworthy addition to the literature on Prisoner’s Dilemma, as it offers new insights and approaches to understanding the properties of memory-one strategies. And it introduces new avenues for future research on a broader spectrum of games and more general strategy spaces. However, there are a few issues that need to be addressed to improve its clarity (especially in the mathematics) and quality before it can be accepted for publication.

Page 2, lines 43-46: It would be clearer if the authors can add references for each of these strategies, separately. For example, the audience may be interested in knowing in which paper ‘random strategy (1/2, 1/2, 1/2, 1/2)’ first appeared.

Page 6, line 171: How do the authors get the two inequalities of p_CC > c/b and p_DD < 1 – c/b?

Page 7, line 224 and Page 13, line 418: There are two different expressions for counting strategies: q = (q_0, q_1, q_2) and q = (q_2, q_1, q_0). The authors need to ensure that the same notations are used consistently throughout.

Page 10, Equations: It is somewhat confusing as the correspondences between the variables (x, y, z, w) and the probabilities (p_CC, p_CD, p_DC, p_DD) change. Could it be better that they are fixed? That is, x always corresponds to p_CC, y to p_CD, so on and so forth.

Page 11, line 269: What is alpha and what is omega? What do these two letters stand for?

Other minor points:

The authors may find it more natural to use “memory-one strategies” instead of “memory-1 strategies” throughout the paper.

For consistency, the two letters C and D (representing two actions in the Prisoner’s Dilemma) need to be either upright font or italic font from the beginning to the end. The authors may take the time to double check.

Reviewer #3: Memory-one strategies are among the best strategy spaces to study direct reciprocity, but their evolutionary dynamics has been difficult to study analytically. This paper provides an analytical description of the above problem under the framework of adaptive dynamics based on the repeated donation game (prisoner’s dilemma). The adaptive dynamics leaves the subspace of counting strategies invariant, and similar invariance occurs in other repeated 2*2 games; Th authors of this paper showed a partial characterization of adaptive dynamics for memory-1 strategies and a full characterization for memory-1 counting strategies. The results are meticulous and interesting, and the conclusions are convincing.

I warmly recommend publication after addressing following comments:

1. In previous works, the reactive strategies, who only depend on the co-player’s previous move, were demonstrated to have some basic properties relative to the memory-one space. It is recommended that the authors include comparative analysis with previous related works in section Results or Discussion.

2. The analytical description of this paper will certainly increase people’s understanding of the properties of memory-one strategies, and it is hoped that similar techniques used in this paper can be used to explore more general strategy spaces, such as including conditional cooperators.

3. It is recommended that the authors appropriately summarize important findings on conditional cooperation strategies in section of Introduction.

**Have the authors made all data and (if applicable) computational code underlying the findings in their manuscript fully available?**

Reviewer #1: None

Reviewer #2: Yes

Reviewer #3: Yes

PLOS authors have the option to publish the peer review history of their article (what does this mean?). If published, this will include your full peer review and any attached files.

Reviewer #1: No

Reviewer #2: No

Reviewer #3: **Yes: **Lei Shi

Figure Files:

Data Requirements:

Reproducibility:

References:

---

## [Decision Letter · Decision Letter 1]

13 Jun 2023

Dear Dr. Hilbe,

We are pleased to inform you that your manuscript 'Adaptive dynamics of memory-1 strategies in the repeated donation game' has been provisionally accepted for publication in PLOS Computational Biology.

Best regards,

Feng Fu

Academic Editor

PLOS Computational Biology

James O'Dwyer

Section Editor

PLOS Computational Biology

Reviewer's Responses to Questions

**Comments to the Authors:**

Reviewer #1: The revised MS seems aufficiently enough to be published...

Reviewer #2: The authors have revised their manuscript according to the comments. I would suggest acceptance of the latest version.

Reviewer #3: Authors have addressed all my comments, and the manuscript can be accepted.

**Have the authors made all data and (if applicable) computational code underlying the findings in their manuscript fully available?**

Reviewer #1: Yes

Reviewer #2: Yes

Reviewer #3: Yes

PLOS authors have the option to publish the peer review history of their article (what does this mean?). If published, this will include your full peer review and any attached files.

Reviewer #1: No

Reviewer #2: No

Reviewer #3: No

---

## [Editor Report · Acceptance letter]

24 Jun 2023

PCOMPBIOL-D-23-00335R1 

Adaptive dynamics of memory-1 strategies in the repeated donation game

Dear Dr Hilbe,

I am pleased to inform you that your manuscript has been formally accepted for publication in PLOS Computational Biology. Your manuscript is now with our production department and you will be notified of the publication date in due course.

With kind regards,

Zsofia Freund
